# MIND Pattern Nutritional Intervention Modulates Mediterranean Diet Adherence and Gut Microbiota in Alzheimer’s Disease: An Observational Case–Control Study

**DOI:** 10.3390/nu18020193

**Published:** 2026-01-07

**Authors:** Laura Di Renzo, Glauco Raffaelli, Barbara Pala, Rossella Cianci, Daniele Peluso, Giovanni Gambassi, Vincenzo Giambra, Antonio Greco, David Della Morte Canosci, Antonino De Lorenzo, Paola Gualtieri

**Affiliations:** 1Section of Clinical Nutrition and Nutrigenomics, Department of Biomedicine and Prevention, Tor Vergata University of Rome, Via Montpellier 1, 00133 Rome, Italy; laura.di.renzo@uniroma2.it (L.D.R.); david.dellamorte@uniroma2.it (D.D.M.C.); delorenzo@uniroma2.it (A.D.L.);; 2PhD School of Applied Medical-Surgical Sciences, Tor Vergata University of Rome, Via Montpellier 1, 00133 Rome, Italy; glauco.raffaelli@yahoo.it (G.R.); daniele.peluso@uniroma2.it (D.P.); 3School of Specialization in Food Science, Tor Vergata University of Rome, Via Montpellier 1, 00133 Rome, Italy; 4Unit of Cardiology, Istituto Dermopatico dell’Immacolata—IRCCS, 00167 Rome, Italy; 5Department of Translational Medicine and Surgery, Catholic University of the Sacred Heart, Fondazione Policlinico Universitario A. Gemelli, IRCCS, 00168 Rome, Italy; giovanni.gambassi@unicatt.it (G.G.); 6Hematopathology Unit, Institute for Stem Cell Biology, Regenerative Medicine and Innovative Therapeutics (ISBReMIT), Fondazione IRCCS Casa Sollievo della Sofferenza, 71013 San Giovanni Rotondo, Italy; v.giambra@operapdrepio.it (V.G.); 7Complex Unit of Geriatrics, Department of Medical Sciences, Fondazione IRCCS Casa Sollievo della Sofferenza, 71013 San Giovanni Rotondo, Italy; a.greco@operapadrepio.it (A.G.)

**Keywords:** Alzheimer’s disease, MEDAS questionnaire, gut microbiota

## Abstract

**Background:** Evidence on non-restrictive MIND pattern interventions in Alzheimer’s (ALZ) disease remains limited. **Methods:** In an observational case–control study, 60 participants (ALZ, n = 30; cognitively healthy controls, n = 30) completed baseline (T0) and follow-up (T1) after structured MIND counseling. Adherence was assessed via the MEDAS questionnaire. Stool samples (16S rRNA profiling) were taken and anthropometry and cognitive/functional measures were recorded at T0/T1. **Results:** In the ALZ group, MEDAS improved as adherence to the Mediterranean diet increased (increasing the use of vegetables ≥ 2/day, *p* < 0.01; and lowering butter adoption ≤ 1/day, *p* = 0.02), with a shift from low to moderate/high adherence; in controls, baseline Mediterranean diet adherence was already high, and changes in MEDAS categories were modest (low adherence from 13.8% to 3.6%, high adherence from 37.9% to 50.0%), with no statistically significant overall change (*p* = 0.39). Regarding gut microbiota (GM), in the ALZ group, alpha diversity increased significantly and Bray–Curtis PCoA separated T0 from T1. Species-level analysis showed increases in SCFA-linked taxa (e.g., *Anaerobutyricum hallii*, *Blautia luti*, *Eubacterium coprostanoligenes*) and reductions in dysbiosis/mucin-degrading taxa (e.g., *Mediterraneibacter torques*, *M. gnavus*, *Agathobacter rectalis*). Between-group Δ(T1 − T0) comparisons at the genus level indicated larger positive shifts in ALZ for *Anaerobutyricum*, *Oscillibacter*, *Faecalicatena*, *Romboutsia*, *Mediterraneibacter*, and *Blautia*, and more negative Δ for *Gemmiger*, *Subdoligranulum*, *Bifidobacterium*, *Clostridium*, and *Collinsella*. sPLS-DA showed partial separation (first two components ≈ 9% variance). **Conclusions:** A structured, non-restrictive MIND intervention was feasible, improved dietary adherence, and accompanied higher diversity and compositional remodeling of the GM in ALZ’s disease. Larger randomized mechanistic studies are warranted.

## 1. Introduction

Defined as the totality of exposures from conception to death, the exposome captures the intricate interplay between the endogenous environment—such as inflammation, oxidative stress, and gut microbiota (GM) composition—and the exogenous environment, including diet, pollutants, socioeconomic status, genetics, and biological responses [1]. This integrative perspective has proven particularly relevant for chronic degenerative conditions, including neurodegenerative diseases such as Alzheimer’s (ALZ) disease, where environmental influences modulate immune, epigenetic, and metabolic pathways over time. Dietary patterns, the GM, and environmental pollutants have all been identified as key exposomic factors capable of driving systemic and neuroinflammation [2].

Combatting ALZ’s disease is an urgent challenge for the foreseeable future, as it represents the most common progressive neurodegenerative disorder, affecting around 30 million people and with incidence rates continually growing [3]. Its etiology is multifactorial, and the pathogenetic mechanisms and causes of progression are far from being clearly understood [4].

Converging epidemiological evidence suggests that a diet rich in plant-based foods and unsaturated fats is associated with slower cognitive decline and a lower risk of ALZ. Of these patterns, the Mediterranean–DASH Intervention for Neurodegenerative Delay (MIND) diet shows the strongest association with reduced cognitive decline in prospective cohorts [5]. Based on the principles of the Mediterranean diet and the Dietary Approaches to Stop Hypertension (DASH) diet, but adapted for neuroprotective purposes, the MIND diet emphasizes foods that have been shown to support brain health and limits those that are associated with vascular and metabolic dysfunction [6].

The exposome framework has increasingly underscored the GM as a pivotal mediator between external factors and host physiology. As a key component of the internal exposome, the GM dynamically interacts with environmental stimuli, influencing immune regulation, metabolic pathways, and neuroinflammatory processes [7,8]. The human GM constitutes a complex and dynamic ecosystem composed of bacteria, viruses, fungi, protozoa, and archaea [9]. The intricate relationship between the GM, immune system, and intestinal mucosa is crucial for the maintenance of health, supporting immune and endocrine functions, energy balance, and nutrition [10]. When this balance is lost, dysbiosis occurs, with evidence of abnormal intestinal barrier permeability and passage of the GM and toxic microbial metabolites through the intestinal mucosa, triggering the onset and development of systemic inflammation [11], subsequent disruption of blood–brain barrier (BBB) integrity, and microglial activation, all of which enhance neuroinflammation and Aβ accumulation [12].

Different studies report discrepancies in GM composition between ALZ patients and healthy controls that may be related to different lifestyles, dietary intake, racial, sexual, and age differences, and environment [13]. However, several studies have associated alterations in GM composition with cognitive impairment [14], suggesting that these alterations can represent an important biomarker in the onset and progression of ALZ’s disease.

Given the lack of data on the effects of a MIND pattern nutritional intervention on Mediterranean diet adherence and GM composition in ALZ’s disease, this study aimed to determine whether a structured MIND program improves Mediterranean diet adherence and induces measurable shifts in microbial diversity and species-level profiles compared with cognitively healthy controls. As secondary aims, we described concurrent anthropometric trajectories and explored associations between dietary adherence and GM changes.

## 2. Materials and Methods

### 2.1. Study Population and Design

This observational case–control study was conducted at the Clinical Nutrition and Nutrigenomics Section, Tor Vergata University of Rome, and at the Fondazione IRCCS “Casa Sollievodella Sofferenza”, San Giovanni Rotondo (FG), Italy, in accordance with the Declaration of Helsinki, Good Clinical Practice, and STROBE guidelines. The protocol received approval from the Ethics Committee of the Calabria Region—Central Area Section (protocol no. 97, dated 20 April 2023) and the Ethics Committee of Fondazione IRCCS “Casa Sollievo della Sofferenza”, Viale Padre Pio, 7, 71013, San Giovanni Rotondo (FG), Italy (Protocol no.272/01/D4 del 12 January 2023). Written informed consent was obtained from each participant; if participants were cognitively incapable of giving consent, informed consent was obtained through the signature of their legal guardian or formally designated caregiver, in accordance with current legislation.

Between July 2023 and April 2024, 60 participants, including patients with ALZ and cognitively healthy controls, completed baseline (T0) and follow-up (T1) assessments after 24 weeks of the intervention. General eligibility required age > 35 years. Participants with a clinical diagnosis of ALZ’s disease were recruited irrespective of age at onset. The diagnosis was established according to the NIA-AA criteria and confirmed by a multidisciplinary team including neurologists and neuropsychologists. As part of the routine diagnostic work-up in those settings, cognitive status had been assessed using the MMSE, the Frontal FAB, and the CDT; for the purposes of the present study, we included only patients whose MMSE score at diagnosis ranged between 15 and 27. These cognitive measures were therefore used exclusively as eligibility criteria and were not collected or analyzed as longitudinal outcomes of the nutritional intervention. Genetic testing for familial ALZ mutations (e.g., APP, PSEN1, PSEN2) was not performed, as no participants exhibited a positive family history or clinical features suggestive of autosomal dominant inheritance. A small subset of participants (n = 3) had onset of symptoms before age 65, but their clinical presentation, dietary response, and microbiota profiles were comparable to older adults participants.

To be enrolled, patients also had to be receiving stable treatment with acetylcholinesterase inhibitors (rivastigmine or donepezil) for at least one month and either be able to provide informed consent or have a legal guardian. Subjects with other forms of dementia, severe comorbidities potentially related to cognitive impairment (e.g., neoplasms, active infections, vitamin B12 deficiency, anemia, thyroid, renal or hepatic diseases), a history of alcohol or substance abuse, head trauma, or other known causes of memory impairment were excluded. Controls (CTRL) were adults without neurodegenerative diseases or exclusion comorbidities.

The nutritional intervention involved scheduled meetings with nutrition professionals who provided structured dietary instructions, sample meal plans, and standardized information material. Reinforcement and adherence checks were scheduled during monthly follow-up. Where cognitive limitations were present, the protocol was explained and shared with the carer or a responsible family member, who received the same operational instructions to ensure its implementation at home. Adherence was monitored by re-administering the MEDAS questionnaire at T0 and T1. This design permitted both between-group comparisons (ALZ vs. CTRL) and an intra-group pre–post analysis of the intervention’s effect.

### 2.2. Data Collection

At baseline, all patients underwent a medical and nutritional assessment, including anthropometric measurements and the collection of fecal samples for analysis of intestinal microbiota composition. Cognitive status was evaluated through the Mini-Mental State Examination (MMSE), the Frontal Assessment Battery (FAB) [15], and the Clock Drawing Test (CDT) [16], following a brief interview with the caregiver. A diagnosis of dementia was supported in all cases by neuro-radiological evidence (computed tomography and/or magnetic resonance imaging). Functional status was measured using the Activities of Daily Living (ADL) [17] and Instrumental Activities of Daily Living (IADL) scales [18].

### 2.3. Anthropometry

According to standardized procedures, anthropometric variables were measured using a height rod and calibrated scales (Invernizzi, Rome, Italy). To minimize possible bias, subjects were assessed in an upright position while wearing only underwear. The accuracy of measurements was set at 0.1 cm for height and 0.1 kg for weight [19].

Body mass index (BMI) was calculated as body weight in kilograms divided by the square of height in meters (kg/m^2^) [20]. To assign a weight category, each BMI value was converted into the ranges established by the World Health Organization: underweight for values below 18.5 kg/m^2^, normal weight for values between 18.5 and 24.9 kg/m^2^, overweight for values between 25.0 and 29.9 kg/m^2^, and obese for values equal to or greater than 30.0 kg/m^2^ [21].

### 2.4. MEDAS

Adherence to the Mediterranean dietary pattern was assessed using the 14-item MEDAS screener. Twelve of the items measure the frequency of consumption of specific food groups, while the remaining two investigate dietary habits consistent with the Mediterranean pattern. Responses are coded as binary (1 for affirmative/compliant response; 0 otherwise) and the total score ranges from 0 to 14. Higher scores indicate greater adherence and are associated with lower cardiovascular risk in the literature, while lower scores denote insufficient adherence. Participants were classified as having low (0–5), moderate (6–10), or high (>10) adherence based on their total score [22].

### 2.5. Stool Sample

Participants were provided with a standardized fecal collection kit, accompanied by detailed written and oral instructions to ensure proper handling and accurate sampling. The procedures were designed to minimize contamination and preserve sample integrity for analysis. The samples were then collected and processed for 16S rRNA gene sequencing. GM profiling was performed by Wellmicro^®^, Bologna (BO), 40138, Italy [23].

### 2.6. Nutritional Intervention (MIND)

The dietary intervention adopted the MIND (Mediterranean–DASH Intervention for Neurodegenerative Delay) pattern and was delivered via individual nutritional counseling at the initial assessment (T0). Practical guidelines, including food lists, sample menus, and meal preparation instructions, were provided directly to participants or, in cases of cognitive impairment, to caregivers/guardians. Reinforcement contacts were arranged during follow-up. No explicit calorie restriction was planned and adherence was monitored by administering the 14-item MEDAS questionnaire at T0 and T1. The rationale behind the MIND approach—a combination of the Mediterranean and DASH models, emphasizing leafy vegetables, berries, extra virgin olive oil, whole grains, legumes, nuts, and fish, while limiting foods high in saturated fat—is supported by recent experimental and observational evidence [6,24,25,26].

Protein intake was standardized at 1.2 g/kg of body weight/day for all participants to ensure adequacy across ages and clinical statuses. This level exceeds the EFSA PRI for healthy adults and aligns with ESPEN guidance for the older adults, falls within the AMDR for protein, and is considered safe in healthy adults without renal disease. In our menus, this target provided ~17% of total energy [27,28,29,30].

Table 1 reports the aggregated nutrient composition of the MIND dietary plans used in the intervention. For each variable, the mean value and standard deviation are presented across menus, with the units indicated. The data were derived from the bromatological analysis and diet plan characteristics tables.

Figure 1 summarizes the eligibility assessment and the subsequent allocation of participants into study groups.

### 2.7. Statistical Analysis

All statistical analyses were performed using R software (version 4.5.1, released on 13 June 2025). Data manipulation and visualization were conducted using the following R packages: dplyr, ggplot2, vegan, Hmisc, openxlsx, pheatmap, and mixOmics.

Intra-group changes over time (T1 vs. T0) were assessed within the ALZ group and the CTRL group using the Wilcoxon signed-rank test (stats::wilcox.test, paired = TRUE), suitable for non-parametric, paired data.

Between-group comparisons of the delta values (Δ = T1 − T0) for microbial relative abundance were carried out using the Wilcoxon rank-sum test (stats::wilcox.test, paired = FALSE). Median differences and log_2_ fold changes were calculated for each taxon. Fold changes were computed with numerical stabilization (pseudo-count = 1 × 10^−6^), and only taxa with |log_2_FC| ≥ 1 and *p*-value < 0.05 were considered significant.

Microbial alpha diversity was estimated using the Shannon and Simpson indices (vegan::diversity). Differences over time were evaluated with the Friedman test (stats::friedman.test) on repeated measures, stratified by individual ID (protocollo).

Beta diversity was assessed using Bray–Curtis dissimilarity (vegan::vegdist) and visualized via Principal Coordinates Analysis (vegan::cmdscale). Group differences were tested using PERMANOVA (vegan::adonis2) and beta-dispersion analysis (vegan::betadisper and vegan::permutest).

To explore associations between microbial delta values and clinical parameter changes (ΔT1 − T0), Spearman’s rank correlations were computed (Hmisc::rcorr). Correlation matrices were visualized using heatmaps (pheatmap::pheatmap), with significance annotations (*p*  < 0.05, * *p*  < 0.01, ** *p*  < 0.001). Multiple testing correction was applied using the Benjamini–Hochberg false discovery rate (stats::*p*.adjust, method = “BH”).

Supervised multivariate analysis was conducted using sparse partial least squares discriminant analysis (sPLS-DA) from the mixOmics package (mixOmics::splsda). The analysis was applied to the delta matrix (Δ microbiota), with group (ALZ vs. CTRL) as the response variable. The first two components (ncomp = 2) were extracted, selecting the top 30 features for each component (keepX = c(30, 30)). Species with the highest loading values were identified and visualized via individual projection plots (mixOmics::plotIndiv) and variable contribution plots (mixOmics::plotVar).

To quantify the biological relevance of between-group Δ(T1 − T0) differences, we calculated effect sizes at the genus level. Wilcoxon’s effect size (r) was computed using rank-biserial correlation (rstatix::wilcox_effsize), and Cliff’s delta was calculated using the effsize::cliff.delta function. FDR-adjusted *p*-values (Benjamini–Hochberg) and effect size magnitudes were reported to aid interpretation.

## 3. Results

### 3.1. Study Participants and Baseline Characteristics

A total of 60 participants, with an overall mean age of 64.73 ± 16.32 years, met the inclusion criteria and completed assessments at baseline (T0) and follow-up (T1). Although the general eligibility criterion allowed for enrolment from 35 years of age, the actual study population predominantly consisted of older adults, with participants ranging from 35 to 85 years and age distributions largely centered in the seventh decade of life. A small subset of participants (n = 3) had onset of symptoms before age 65, but their clinical presentation, dietary response, and gut microbiota profiles were comparable to older adults.

Baseline comparisons between groups were performed using independent-samples *t*-tests for continuous variables and Fisher’s exact test for categorical variables (Table 2).

For consistency across tables and figures, subgroup labels were harmonized as ALZ-T0/ALZ-T1 for the Alzheimer’s cohort at baseline and follow-up, and CTRL-T0/CTRL-T1 for controls; T0 denotes baseline and T1 denotes follow-up. At baseline, the ALZ cohort (16 women, 53.3%; 14 men, 46.7%) had a mean body weight of 69.25 ± 13.01 kg and BMI 28.31 ± 4.60 kg/m^2^. At T1, neither body weight (68.81 ± 12.26 kg) nor BMI (28.26 ± 4.48 kg/m^2^) showed statistically significant differences (Table 3). The control group exhibited a mean weight of 76.12 ± 19.93 kg and BMI 25.78 ± 5.54 kg/m^2^. At T1, a modest decline was observed in weight (75.19 ± 18.66 kg) and BMI (25.46 ± 5.03 kg/m^2^), again without statistical significance (Table 4).

### 3.2. Adherence to the Mediterranean Diet Following the MIND Model-Based Intervention

Following administration of the MEDAS questionnaire at baseline, it was repeated at T1. The second administration showed greater adherence to the Mediterranean model.

In the ALZ group (Table 5), the proportion of participants meeting the vegetable intake criterion increased from 23.30% at T0 to 70.00% at T1 (+46.70 percentage points; absolute change, *p* = 0.00). Adherence to the criterion of consuming ≤1 serving of butter per day increased from 73.30% to 96.70% (+23.4 percentage points; *p* = 0.03). For other MEDAS items, the absolute changes suggested a reduction in red meat and sweet consumption and an increase in extra virgin olive oil use, though these differences were not statistically significant (all *p* > 0.05; Table 5). Overall MEDAS adherence shifted accordingly: low adherence decreased from 33.3% to 13.3%, while moderate and high adherence increased from 53.3% to 70.0% and from 13.30% to 16.70%, respectively. Although the proportion of participants classified as low adherence decreased and moderate/high adherence increased, the overall change in MEDAS categories was not statistically significant (*p* = 0.25).

Adherence distribution in the CTRL group (Table 6) improved overall, with a decrease in low adherence (from 13.80% to 3.60%) and an increase in high adherence (from 37.90% to 50.00%), without a statistically significant change in category distribution (*p* = 0.39).

Figure 2 and Figure 3 show the radar chart of affirmative responses to the 14 MEDAS items and the bar chart of overall adherence for each group at T0 and T1, respectively.

### 3.3. Effect of MIND Intervention on Gut Microbiota Composition

The overall profile of the GM, with the list of species detected and their respective mean abundances and standard deviations, according to fecal samples collected at T0 and T1 in the ALZ and CTRL groups, is shown in Appendix A. Table 7 shows the species that were statistically significant in the ALZ group.

*Anaerobutyricum hallii* (from 0.22 ± 0.52 to 2.75 ± 2.47; *p* < 0.01), *Blautia luti* (from 0.62 ± 1.27 to 5.62 ± 4.32; *p* < 0.01), *Blautia glucerasea* (from 0.24 ± 0.57 to 2.25 ± 2.28; *p* < 0.01), *Romboutsia timonensis* (from 0.18 ± 0.67 to 2.05 ± 3.39; *p* < 0.01), *Eubacterium coprostanoligenes* (from 0.18 ± 0.41 to 1.20 ± 0.89; *p* < 0.01), *Faecalicatena contorta* (from 0.09 ± 0.22 to 0.54 ± 0.50; *p* < 0.01), and *Ruminococcus callidus* (from 0.03 ± 0.12 to 0.70 ± 1.29; *p* < 0.01) were observed to increase.

At the same time, there were significant reductions in *Mediterraneibacter torques* (from 1.44 ± 2.28 to 0; *p* = 0.001), *Mediterraneibacter gnavus* (from 0.21 ± 0.55 to 0; *p* < 0.01), *Agathobacter rectalis* (from 1.72 ± 2.70 to 0; *p* < 0.01), and *Ruminococcus bicirculans* (from 0.23 ± 0.43 to 0; *p* < 0.01).

Figure 4 illustrates the relative abundance of species that exhibited statistically significant differences in the Alzheimer’s group.

In the ALZ group, analysis of the microbiota between T0 and T1 showed significant higher medians of the Shannon and Simpson diversity indices at T1 (*p* < 0.001). A richer and more evenly distributed community was revealed (Figure 5).

PCoA on the Bray–Curtis distance showed a separation of communities between the two time points, consistent with a post-intervention compositional rearrangement. Overall, the evidence indicates that the intervention promoted increased diversity and compositional rebalancing of the GM in Alzheimer’s subjects, potentially contributing to improved intestinal and systemic health (Figure 6).

Fold change analysis at the species level identified a set of taxa significantly modulated by the intervention (Figure 7). Several short-chain fatty acid-producing species belonging to the genera *Anaerobutyricum*, *Faecalicatena*, *Mediterraneibacter*, *Romboutsia*, and *Oscillibacter* showed increased relative abundance at T1 compared with T0, whereas species within *Gemmiger*, *Subdoligranulum*, *Bifidobacterium*, *Clostridium,* and *Collinsella* decreased (Figure 7).

In the control group, the comparison between T0 (CTRL-T0) and T1 (CTRL-T1) did not reveal any statistically significant differences in the composition of the microbiota (Table 8). Only minor variations in mean abundances were found: decreases for *Faecalibacterium prausnitzii* (from 10.79 ± 5.39 to 9.04 ± 6.25), *Fusicatenibacter saccharivorans* (from 2.26 ± 2.05 to 1.70 ± 1.80), and *Roseburia faecis* (from 2.12 ± 2.51 to 1.58 ± 1.99); slight increases for *Blautia obeum* (from 0.64 ± 0.88 to 1.48 ± 2.54) and *Oscillibacter valericigenes* (from 1.06 ± 1.60 to 1.19 ± 2.06); further reductions for *Sporobacter thermitidis* (from 0.46 ± 0.81 to 0.22 ± 0.43) and *Roseburia hominis* (from 0.53 ± 0.65 to 0.24 ± 0.39).

Figure 8 illustrates the percentage abundance of certain species in the control group at T0 and T1. While these values are not statistically significant, they have been included to provide a more complete description and additional evidence of compositional stability.

In the control group, alpha diversity assessed using Shannon and Simpson indices showed no significant differences between baseline (CTRL-T0) and post-intervention (CTRL-T1); the medians were comparable and the Wilcoxon test confirmed the absence of statistically significant variations (Figure 9).

PCoA on the Bray–Curtis distance showed extensive overlap between T0 and T1 samples, with no clear temporal separation and no significant differences (Figure 10).

At the species level, fold change analysis did not identify taxa with statistically significant variations between T0 and T1; the fluctuations observed fell within inter-individual variability.

### 3.4. Analysis of the Differences in Microbiota Abundance Between ALZ and CTRL

The subject-level change in relative abundance (Δ = T1 − T0) was computed for each genus, and the Wilcoxon rank-sum test was used to test for between-group differences in the median Δ (Table 9).

Significant genera with a greater positive median Δ(T1 − T0) in ALZ than in CTRL were as follows: *Anaerobutyricum* (median Δ: ALZ 2.20 vs. CTRL 0.00; *p* < 0.01), *Romboutsia* (0.25 vs. 0.00; *p* < 0.01), *Faecalicatena* (0.40 vs. 0.00; *p* < 0.01), *Mediterraneibacter* (0.15 vs. 0.00; *p* < 0.01), *Oscillibacter* (1.25 vs. 0.00; *p* = 0.02), and *Blautia* (3.75 vs. 0.15; *p* = 0.04).

Conversely, lower median Δ(T1 − T0) values were found in ALZ than in CTRL for the following: *Gemmiger* (0.05 vs. 0.00; *p* < 0.01), *Subdoligranulum* (0.35 vs. 0.00; *p* < 0.01), *Bifidobacterium* (1.95 vs. −0.35; *p* = 0.04), *Clostridium* (0.85 vs. 0.00; *p* = 0.04), and *Collinsella* (0.80 vs. 0.00; *p* = 0.04).

Additional genera showing significant between-group differences included the following: *Arthrobacter* (*p* < 0.01), *Sporobacter* (*p* < 0.01), *Erysipelatoclostridium* (*p* = 0.01), *Veillonella* (*p* = 0.04), *Escherichia* (*p* = 0.05), *Lachnospira* (*p* = 0.01), *Holdemanella* (*p* = 0.04), *Limosilactobacillus* (*p* = 0.04), *Christensenella* (*p* = 0.01), *Methanobrevibacter* (*p* = 0.02), and *Actinomyces* (*p* = 0.02).

Figure 11 shows the distribution of Δ(T1 − T0) for the aforementioned taxa.

Further comparisons between groups were conducted by calculating the log_2_ fold change (log_2_FC) between the medians of the Δ(T1 − T0) relative abundances, with Wilcoxon rank-sum tests used to assess statistical significance. Genera were deemed differentially varying when *p* < 0.05 and |log_2_FC| ≥ 1. To complement statistical significance, effect sizes were computed for each genus. Wilcoxon test correlation and Cliff’s delta consistently indicated moderate-to-large effects for genera showing significant between-group differences, supporting the robustness of the observed directional changes.

Seven genera met these criteria (Table 10): *Anaerobutyricum* (median Δ ALZ: 2.20 vs. CTRL: 0.00; log_2_FC: 21.07; *p* < 0.01), *Oscillibacter* (1.25 vs. 0.00; log_2_FC: 20.25; *p* = 0.02), *Faecalicatena* (0.40 vs. 0.00; log_2_FC: 18.61; *p* < 0.01), *Romboutsia* (0.25 vs. 0.00; log_2_FC: 17.93; *p* < 0.01), *Mediterraneibacter* (0.15 vs. 0.00; log_2_FC: 17.19; *p* < 0.01), *Blautia* (3.75 vs. 0.15; log_2_FC: 4.64; *p* = 0.04), and *Bifidobacterium* (−1.95 vs. − 0.35; log_2_FC 2.48; *p* = 0.04). For *Bifidobacterium*, the median Δ was more negative in ALZ than in CTRL.

Figure 12 shows the log_2_ fold changes (ALZ vs. CTRL), calculated according to the medians of Δ(T1 − T0), for genera with significant differences.

Supervised sPLS-DA was performed on the Δ(T1 − T0) of the relative abundance of genera for the ALZ vs. CTRL comparison. The score plot (Figure 13) shows partial separation between the groups along Component 1. The first two components explained 4% and 5% of the variance, respectively, for a total of 9%. In the loading/variable projection plot (Figure 14), the genera contributing most to group separation were predominantly Bacillota, with *Anaerobutyricum*, *Oscillibacter*, *Romboutsia,* and *Blautia* showing the largest absolute loadings on Components 1–2. *Pseudomonadota* also contributed, notably *Klebsiella* and *Shigella*. Additional genera with non-trivial loadings included *Subdoligranulum*, *Gemmiger*, *Holdemanella*, *Christensenella,* and *Bifidobacterium*. In this representation, greater distance from the origin denotes a stronger discriminative contribution to the model.

Model validation revealed that, despite the low total variance explained by the first two components (≈9%), the sPLS-DA model achieved stable classification performance. Cross-validation using 50 repetitions of 5-fold resampling yielded a consistent Balanced Error Rate (BER) of ~0.21 ± 0.03 for both components. In addition, Q^2^ total values were ~0.12 for Component 1 and ~0.11 for Component 2, suggesting modest but non-negligible predictive performance. While formal per-mutation testing was not conducted within the mixOmics framework, the repeated cross-validation serves as a reliable surrogate for model robustness. Furthermore, stability analysis showed that genera such as *Anaerobutyricum*, *Gemmiger*, and Bifidobacterium were consistently selected across resamples, supporting their biological relevance in group discrimination.

## 4. Discussion

Our findings indicate that adoption of the MIND dietary pattern produced measurable and clinically relevant changes in both food choices and gut microbial ecology in patients with ALZ. After six months, the proportion of participants meeting the MEDAS vegetable criterion significantly increased, the number of participants with butter intake ≤ 1 serving/day significantly rose, and overall adherence shifted from low to predominantly moderate/high levels. These improvements in diet quality, achieved in a cognitively impaired population, support the feasibility of targeting nutritional patterns as a modifiable factor in ALZ’s disease and are consistent with prior evidence linking MIND adherence to slower cognitive decline and reduced dementia risk. The observed shift from low to at least moderate adherence, even if not statistically significant, indicates a qualitative transition towards higher consumption of plant-based foods and extra virgin olive oil and lower intake of saturated fats; in longitudinal Mediterranean and MIND diet cohorts, similar improvements in adherence have been associated with reduced cardiometabolic risk and slower cognitive decline, suggesting that the MEDAS changes observed in our ALZ patients are clinically relevant even though the present study was not powered to detect a hard clinical endpoint. In parallel with the dietary changes, the intervention produced substantial remodeling of the GM. Taxa recognized for their ability to synthesize SCFAs, such as *Anaerobutyricum hallii*, *Blautia luti*, *Blautia glucerasea*, *Romboutsia timonensis*, *Eubacterium coprostanoligenes*, *Faecalicatena contorta*, and *Ruminococcus callidus*, were enriched after the intervention. These bacteria contribute to butyrate, acetate, and propionate biosynthesis, metabolites that sustain epithelial barrier function, provide energy to colonocytes, and influence systemic and neural immune signaling. Recent mechanistic studies suggest that SCFAs can modulate microglial reactivity, synaptic transmission, and neuroinflammatory cascades, thereby linking intestinal metabolism to brain homeostasis [31]. The statistically significant increase in these beneficial taxa in ALZ participants suggests that the MIND dietary pattern can restore a more protective microbial milieu and enhance metabolic functions relevant for neuroprotection. Conversely, we documented a reduction in microbial groups associated with dysbiosis and pro-inflammatory potential, including *Anaerobutyricum hallii*, *Mediterraneibacter torques*, *Mediterraneibacter gnavus*, *Agathobacter rectalis*, and *Ruminococcus bicirculans.* These taxa have previously been implicated in the degradation of mucins, the weakening of epithelial barrier integrity, and the release of inflammatory compounds. For instance, *Mediterraneibacter gnavus* produces specific polysaccharides that activate immune pathways and contribute to intestinal inflammation [32]. *Mediterraneibacter torques* has also been linked to unfavorable mucosal states [33], while a higher abundance of *Agathobacter rectalis* has been reported in dysbiotic communities of older adults and cognitively impaired populations [34]. The depletion of these taxa in the ALZ cohort after the intervention is therefore consistent with a shift toward an intestinal configuration less prone to pro-inflammatory activity. In addition to taxonomic shifts, ecological indices provided further evidence of microbial restructuring. Alpha diversity, assessed by Shannon and Simpson indices, significantly increased following MIND (Figure 5), underlining higher richness and evenness of microbial communities after the dietary intervention. Moreover, beta diversity analysis demonstrated significant clustering of microbial populations in PCoA plots pre- versus post-intervention, confirming that the diet drove a distinct reorganization of the gut ecosystem. Reduced microbial diversity is frequently reported as a characteristic of ALZ-associated dysbiosis, often linked to heightened inflammation and altered signaling along the gut–brain axis [35]. Between-group analyses of genus-level Δ(T1 − T0) relative abundance showed significantly larger increases in ALZ than in controls for *Anaerobutyricum*, *Oscillibacter*, *Faecalicatena*, *Romboutsia*, *Mediterraneibacter,* and *Blautia*, whereas more negative Δ(T1 − T0) values in ALZ were observed for *Gemmiger*, *Subdoligranulum*, *Bifidobacterium*, *Clostridium,* and *Collinsella*. The decrease in *Bifidobacterium*, despite its recognized beneficial role, should be interpreted with caution, as genus-level 16S profiles do not distinguish between species with potentially divergent functions, and relative abundances are influenced by the expansion of other SCFA-producing taxa. In this context, the observed pattern may reflect compositional rebalancing within a broader saccharolytic and SCFA-producing community rather than a detrimental loss of *bifidobacteria*. The predominance of *Firmicutes* among genera with positive Δ(T1 − T0) is compatible with a fiber-responsive, saccharolytic shift in the ecosystem: *Lachnospiraceae* and related taxa are enriched under higher fiber intakes and generate SCFAs through carbohydrate fermentation [36,37,38]. Moreover, the current literature suggests that *Bifidobacterium* dynamics in the older adults are complex: while the genus contributes to longevity mechanisms, its abundance physiologically declines with aging and is influenced by long-term lifestyle and metabolic factors. Comparative studies on healthy aging show that such declines often occur without adverse functional consequences, reflecting shifts toward other butyrate-producing species that maintain gut homeostasis [39,40]. In line with this interpretation, *Romboutsia* and *Oscillibacter* have experimental and culture-based evidence for carbohydrate utilization and the production of SCFAs or related metabolites, although the net health associations of *Oscillibacter* appear context-dependent across settings [41,42]. Moreover, large human cohorts underscore that microbial compositional shifts track host lipid and sterol metabolism, highlighting the broader metabolic plasticity of commensals relevant to systemic axes [43]. In our sPLS-DA model at the gender level, the score plot showed only partial separation between ALZ and CTRL, with the first components explaining a modest amount of variance (4–5%). This is consistent with the use of supervised methods such as sPLS-DA, which maximize separation between classes, but require rigorous stability and validity assessments (e.g., repeated cross-validation and permutation tests) to avoid overfitting and the excessive interpretation of latent patterns, particularly when the explained variance is low. While group separation was modest, the sPLS-DA model exhibited stable classification performance, as supported by cross-validation and Q^2^ metrics, suggesting the presence of subtle but biologically meaningful differences. Furthermore, since microbiome data are compositional, the selection of variables and the interpretation of loadings should consider the appropriate transformations and approaches for compositional data. In this context, the higher loadings observed for *Bacillota* genera (*Anaerobutyricum*, *Oscillibacter*, *Romboutsia* and *Blautia*) and some *Pseudomonadota (Klebsiella* and *Shigella)* suggest a contribution to discrimination in our dataset. However, these signals should be considered exploratory and integrated with univariate results and independent validations [44]. Thus, our findings suggest that MIND adherence restores not only the taxonomic balance but also the ecological stability of the intestinal environment. These results align with the growing body of evidence emphasizing the role of diet in shaping the microbiota–brain connection. Mediterranean-like dietary patterns have been shown to promote greater microbial diversity, enrich SCFA-producing organisms, and attenuate systemic inflammation [45]. Mendelian randomization analyses have even demonstrated causal associations between specific microbial taxa and ALZ susceptibility [46]. Furthermore, dietary interventions and microbiota-targeted approaches such as probiotics and prebiotics have shown promise in modulating cognitive trajectories, though results remain heterogeneous [47]. In this context, our findings contribute novel evidence supporting nutritional modulation as a feasible and effective strategy in ALZ management. The strengths of the present study include the integration of validated dietary assessment tools with detailed microbiota profiling, as well as the focus on a clinically vulnerable group with established ALZ. The demonstration of significant dietary and microbial changes after a relatively short intervention underscores the responsiveness of both diet and the GM in neurodegenerative conditions. However, several limitations must be acknowledged. A limitation of the present study is the significant age difference between the two groups. Age mismatch between cases and controls represents a known limitation in microbiome studies on neurodegenerative disorders. Notably, a recent systematic review by Heravi et al. [48] highlighted that some studies investigating gut microbiota alterations in Parkinson’s and ALZ’s disease included control groups that were not age-matched. For example, Vascellari et al. [49] reported a significant age difference between patients and controls, with mean ages of 71.39 ± 10.99 and 51.67 ± 12.42 years, respectively. Similarly, other influential studies in the field have reported significant age differences between cases and healthy controls [14,50] (*p* < 0.01). These examples indicate that, while age matching is desirable, it is not always feasible in clinical microbiome research and does not preclude the identification of biologically meaningful disease-associated microbial signatures. This imbalance reflects the pragmatic recruitment strategy adopted for the control group, which consisted of individuals continuously followed at our centers. Controls were selected to allow for rapid enrollment of participants willing to undergo all required assessments, enabling data collection to be performed in parallel with the patient group and under comparable conditions and closely matched sampling periods. Although the groups are not fully comparable in terms of age, this limitation should be considered acceptable within the context of the study design. The exploratory nature of microbiome multivariate modeling and the relatively small sample size require cautious interpretation of effect sizes and clustering patterns. The duration of the trial could be insufficient to assess the sustainability of the observed changes or to evaluate long-term cognitive outcomes. This limitation is amplified in ALZ, where many participants may be unable to maintain adequate oral intake, may no longer remain oriented to reality or capable of consistent participation, and may not survive long enough for extended outcome assessment. Moreover, the lack of functional metagenomic and metabolomic analyses prevented direct confirmation of SCFA levels and microbial metabolic activity. Dietary adherence was assessed using the MEDAS questionnaire administered by trained dietitians with the support of caregivers, which likely improved the internal consistency of the data; however, self-reported measures remain prone to recall and social desirability bias, particularly in cognitively impaired individuals. These aspects should be addressed in future multicenter and longer-term trials.

## 5. Conclusions

This observational case–control study found that a structured MIND pattern intervention, delivered without explicit caloric restriction, was feasible for patients with AD and produced measurable effects on both diet quality and gut microbial ecology. In the ALZ cohort, adherence to Mediterranean-like dietary targets increased and the gut microbiota exhibited greater alpha diversity and a distinct beta diversity profile compared to baseline. There were also shifts at the genus and species levels that were consistent with enhanced fermentative capacity and marked increases in SCFA-linked taxa, such as *Anaerobutyricum*, *Blautia*, *Romboutsia*, *Faecalicatena*, *Mediterraneibacter* and *Oscillibacter*. Concomitant reductions were observed in taxa that are often associated with dysbiosis or mucin degradation, including *Mediterraneibacter torques*, *R. gnavus* and *Agathobacter rectalis*. Between-group Δ(T1 − T0) analyses corroborated these patterns, whereas the controls remained largely stable. These findings support the use of dietary modulation as a pragmatic, caregiver-compatible strategy to influence the gut–brain axis in ALZ’s disease. They also encourage further research in larger and more diverse cohorts, ideally using randomized designs and extended follow-up. This research should integrate metagenomics, metabolomics, markers of barrier integrity and neuroinflammation, and standardized cognitive and functional endpoints.

## Figures and Tables

**Figure 1 nutrients-18-00193-f001:**
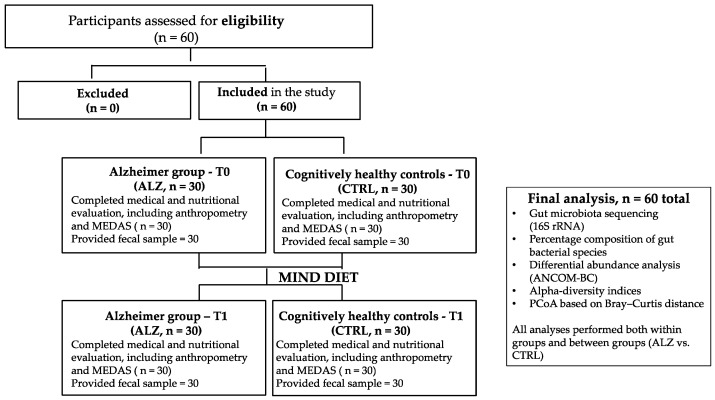
An overview of the eligibility assessment, inclusion of participants, and distribution into Alzheimer (ALZ) and control groups (CTRL) at T0 and T1.

**Figure 2 nutrients-18-00193-f002:**
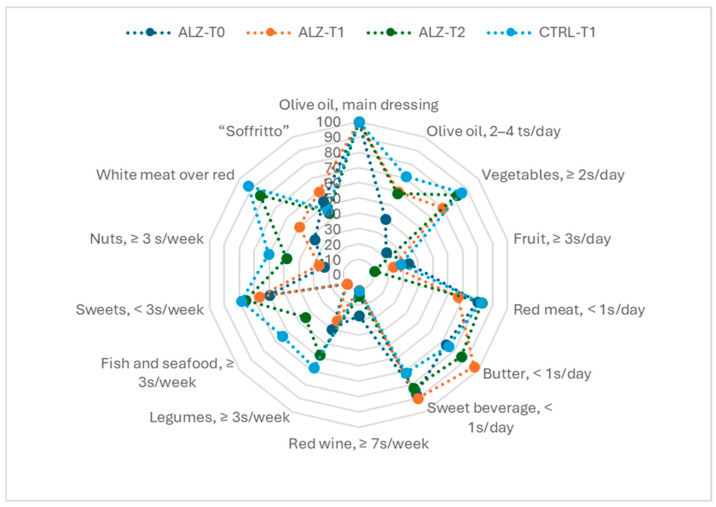
A radar plot of the percentage of participants meeting each MEDAS item criterion at baseline and follow-up in Alzheimer’s and control groups. Each spoke corresponds to one of the 14 MEDAS items; lines represent subgroup profiles, with higher values indicating greater item-level adherence.

**Figure 3 nutrients-18-00193-f003:**
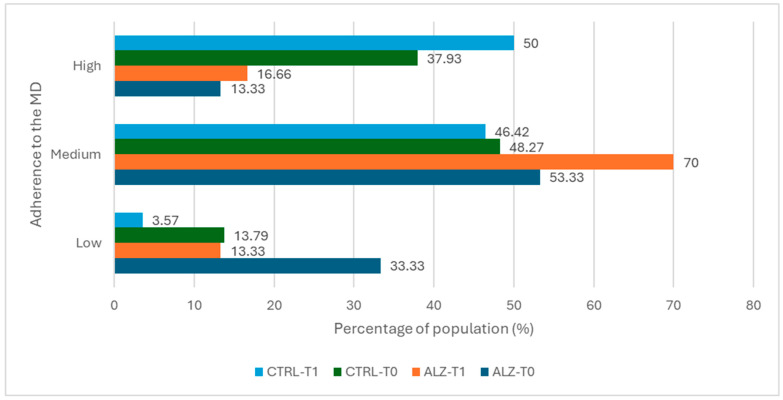
Distribution (%) of Mediterranean diet adherence categories (low, moderate, high; derived from MEDAS total score) in Alzheimer’s and control groups at baseline and follow-up. Bars represent proportion of participants within each category.

**Figure 4 nutrients-18-00193-f004:**
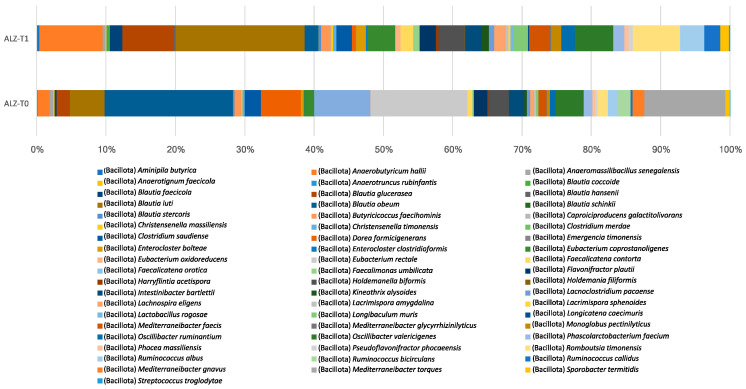
Species-level profiles of gut microbiota in the Alzheimer’s group before (ALZ-T0) and after (ALZ-T1) the nutritional intervention. The stacked bars show the mean relative abundances (%); labeled species are those showing significant intra-group differences between T0 and T1 according to Wilcoxon signed-rank tests (*p* < 0.05).

**Figure 5 nutrients-18-00193-f005:**
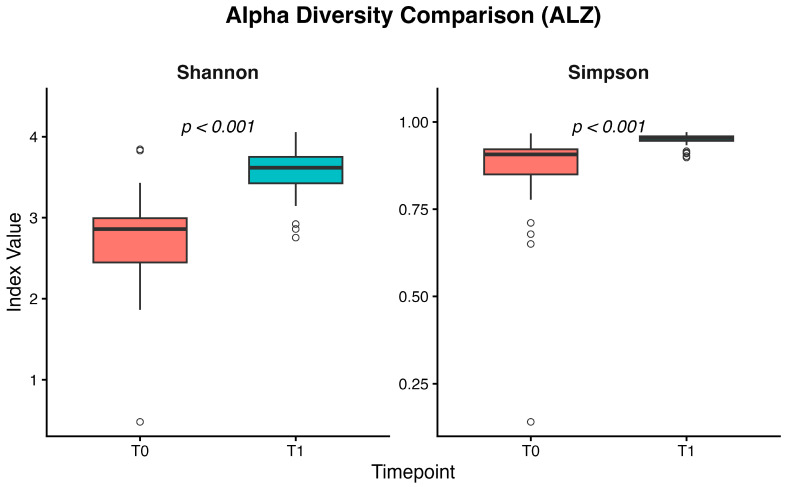
Boxplots of alpha diversity indices (Shannon and Simpson) in the Alzheimer’s group, showing an increase in microbial richness and evenness from T0 to T1. Significant *p*-values (reported above the boxplots) were obtained with Wilcoxon signed-rank tests for paired comparisons between T0 and T1.

**Figure 6 nutrients-18-00193-f006:**
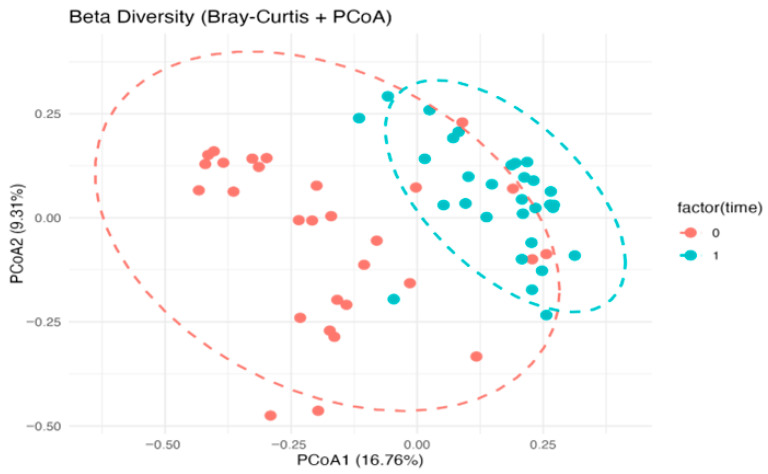
PCoA of gut microbiota based on Bray–Curtis distances in the Alzheimer’s group, showing pre- (T0, red) and post-intervention (T1, light blue) samples with 95% confidence ellipses. Differences between T0 and T1 were tested by PERMANOVA on Bray–Curtis distances, with homogeneity of dispersion assessed by beta-dispersion.

**Figure 7 nutrients-18-00193-f007:**
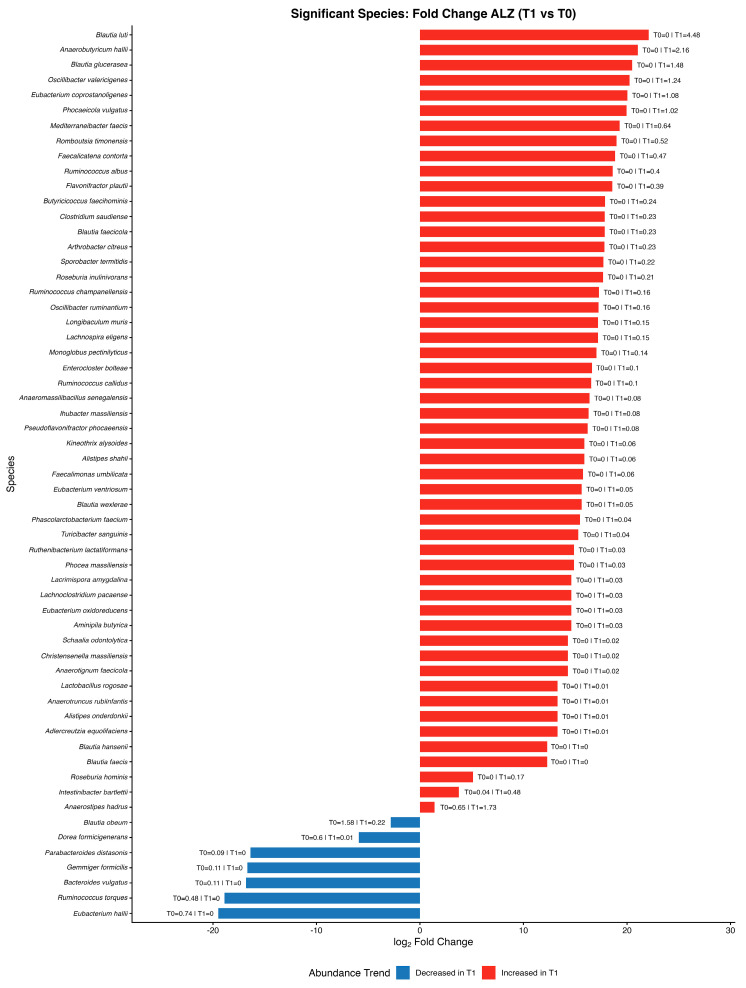
A bar plot of the log_2_ fold change (T1 vs. T0) in species-level abundances in the Alzheimer’s group. Red bars indicate the taxa that increased at T1 and blue bars indicate those that decreased at T1. Only species with statistically significant changes in paired comparisons between T0 and T1 (Wilcoxon signed-rank test, |log_2_FC| ≥ 1, *p* < 0.05) are shown.

**Figure 8 nutrients-18-00193-f008:**
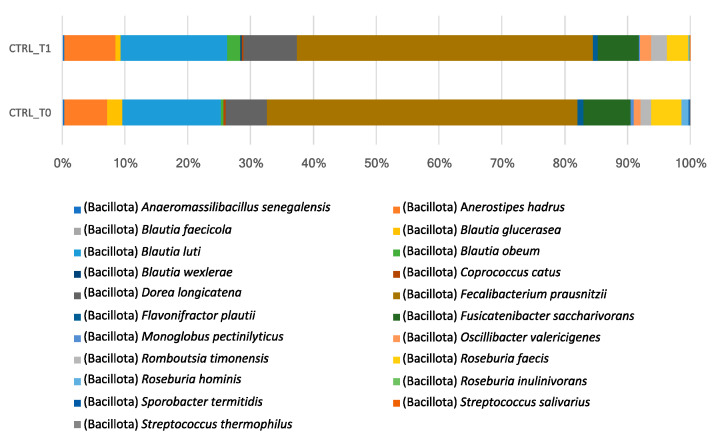
Cumulative relative abundances (%) of the main bacterial species in the control group before (CTRL-T0) and after (CTRL-T1) the nutritional intervention. Differences between CTRL-T0 and CTRL-T1 were assessed using Wilcoxon signed-rank tests; none of the tested species reached statistical significance (all *p* > 0.05).

**Figure 9 nutrients-18-00193-f009:**
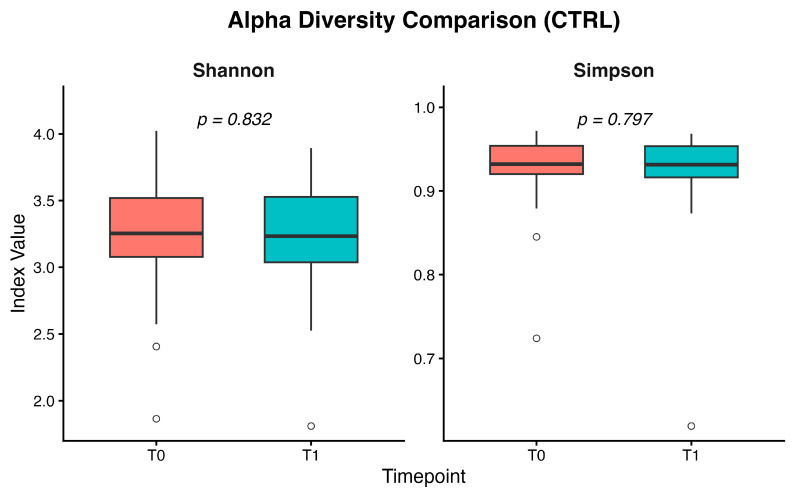
Boxplots of Shannon and Simpson alpha diversity indices in the control group at baseline (T0) and after the intervention (T1). Medians largely overlap, and Wilcoxon signed-rank tests showed no significant differences between time points (Shannon *p* = 0.83; Simpson *p* = 0.80).

**Figure 10 nutrients-18-00193-f010:**
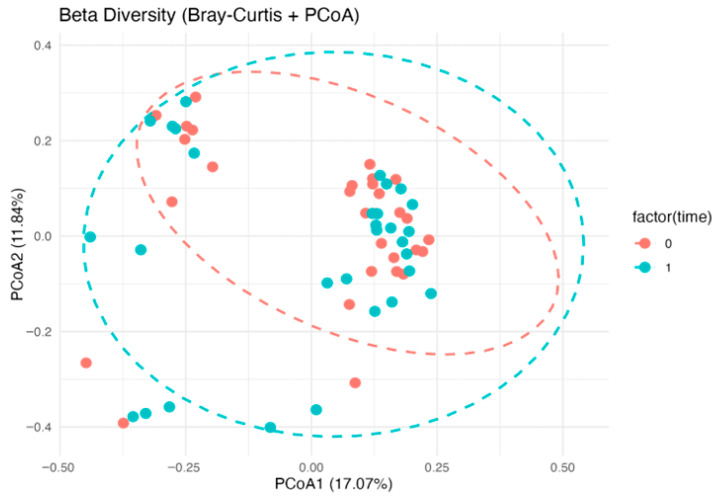
PCoA of gut microbiota based on Bray–Curtis distances in the control group at baseline (T0, red) and after the intervention (T1, light blue), with 95% confidence ellipses. PERMANOVA on Bray–Curtis distances showed no significant differences between time points, and beta-dispersion tests indicated homogeneous dispersion.

**Figure 11 nutrients-18-00193-f011:**
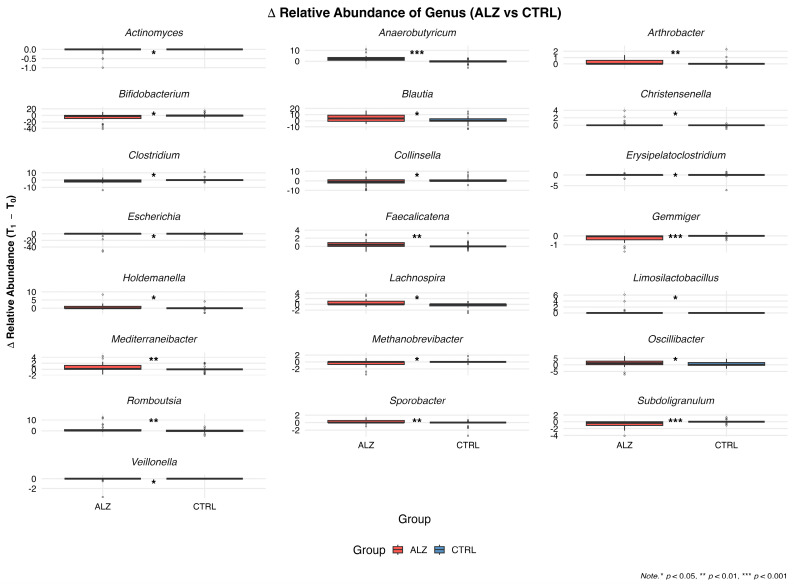
Boxplots of changes (ΔT1 − T0) in genus-level relative abundance in Alzheimer’s (ALZ) and control (CTRL) groups. Between-group differences in Δ were tested with Wilcoxon rank-sum tests with Benjamini–Hochberg FDR correction; asterisks indicate genera with significant ALZ–CTRL differences.

**Figure 12 nutrients-18-00193-f012:**
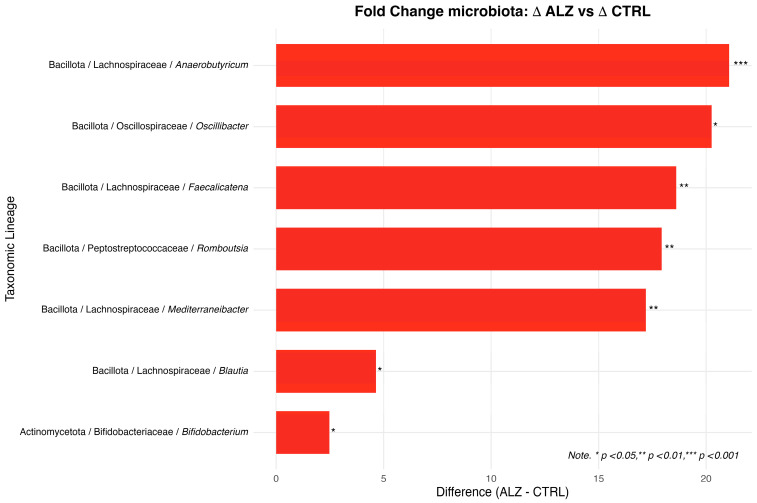
Bar chart of log_2_ fold change (ALZ vs. CTRL) calculated according to median Δ(T1 − T0) for genera with *p* < 0.05 and |log_2_FC| ≥ 1. Effect size is represented by bar length; asterisks indicate genera with significant ALZ–CTRL differences in Δ according to Wilcoxon rank-sum tests (FDR-corrected).

**Figure 13 nutrients-18-00193-f013:**
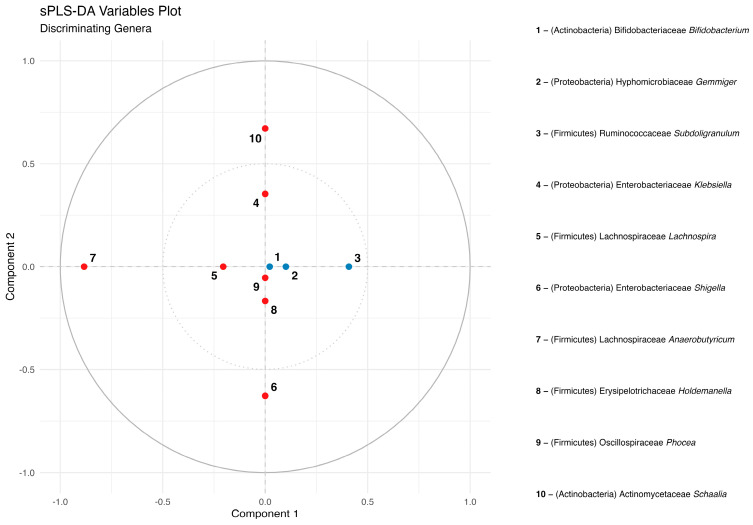
A correlation circle plot from the sparse partial least squares discriminant analysis (sPLS-DA) performed on genus-level Δ(T1 − T0) relative abundances. Each label corresponds to a bacterial genus, positioned according to its loadings on Component 1 and Component 2, which explain 4% and 5% of the variance, respectively; genera located farther from the origin contribute more strongly to the discrimination between ALZ and CTRL groups. Point colors indicate the direction of the contribution to Component 1: blue points represent a positive correlation, while red points represent a negative correlation.

**Figure 14 nutrients-18-00193-f014:**
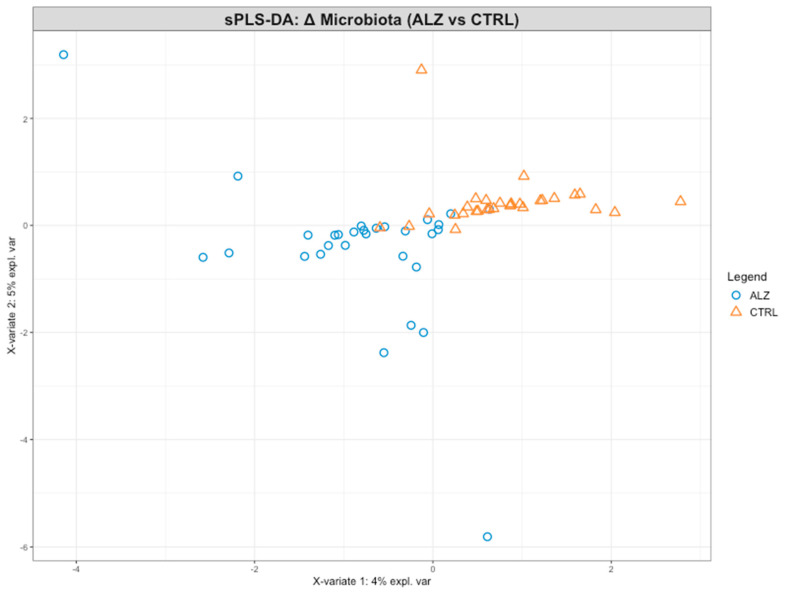
sPLS-DA variable/loading projection plot showing the genera with the greatest contribution to the first two components, labeled by phylum, family, and genus.

**Table 1 nutrients-18-00193-t001:** The nutrient composition of the MIND intervention menus is shown below as the mean ± SD across the various dietary plans.

Variable	Unit	Mean ± SD
Energy	kcal/day	1815.50 ± 191.50
Protein	g/day	75.30 ± 13.60
Protein (% energy)	%	17.25 ± 0.96
Animal protein	g/day	27.50 ± 7.05
Plant protein	g/day	36.25 ± 7.13
Carbohydrate	g/day	208.75 ± 22.93
Carbohydrate (% energy)	%	45.20 ± 0.40
Soluble sugars	g/day	48.00 ± 1.63
Starch	g/day	86.25 ± 29.79
Total fiber	g/day	39.00 ± 3.83
Fat	g/day	69.40 ± 5.31
Fat (% energy)	%	34.90 ± 0.66
Cholesterol	mg/day	179.00 ± 44.40
Saturated fat	g/day	11.18 ± 0.89
Saturated fat (% energy)	%	5.68 ± 0.47
Monounsaturated fat	g/day	41.25 ± 2.87
Polyunsaturated fat	g/day	11.25 ± 0.96
Omega-3	g/day	2.23 ± 0.21
Omega-6	g/day	8.98 ± 0.71
Salt	g/day	1.86 ± 0.09
Sodium	mg/day	632.73 ± 152.27
Glycemic index	AU	45.50 ± 0.00
Mediterranean Adequacy Index (MAI)	AU	10.00 ± 0.00
Atherogenic Index (AI)	AU	0.05 ± 0.00
Thrombogenic Index (TI)	AU	0.20 ± 0.00
ORAC	AU	15,930.00 ± 500.00
PRAL	mEq/day	−16.46 ± 4.50
Omega-6/Omega-3 ratio	ratio	4.22 ± 0.52

**Table 2 nutrients-18-00193-t002:** General and anthropometric characteristics at T0 of ALZ and CTRL groups. ** *p*-value < 0.01.

Variables	ALZ-T0	CTRL-T0	*p*-Value
Sample size	30 (100.00)	30 (100.00)	
Female (%)	16 (53.30)	16 (53.30)	
Male (%)	14 (46.70)	14 (46.70)	
Weight (kg)	69.25 ± 13.01	76.12 ± 19.93	0.12
BMI (kg/m^2^)	28.31 ± 4.6	25.78 ± 5.54	0.07
Age (years)	77.02 ± 7.38	52.54 ± 13.27	0.01 **

Values are expressed as mean ± standard deviation (M ± SD) for continuous variables, and as number and percentage (n (%)) for categorical variables. Abbreviations: BMI (body mass index).

**Table 3 nutrients-18-00193-t003:** General and anthropometric characteristics of ALZ group at T0 vs. T1.

Variables	ALZ-T0	ALZ-T1	*p*-Value
Sample size	30 (100.00)	30 (100.00)	
Female (%)	16 (53.30)	16 (53.30)	
Male (%)	14 (46.70)	14 (46.70)	
Weight (kg)	69.25 ± 13.01	68.81 ± 12.26	0.93
BMI (kg/m^2^)	28.31 ± 4.60	28.26 ± 4.48	0.98

Values are expressed as mean ± standard deviation (M ± SD) for continuous variables, and as number and percentage (n (%)) for categorical variables. Abbreviations: BMI (body mass index).

**Table 4 nutrients-18-00193-t004:** General and anthropometric characteristics of CTRL group at T0 vs. T1.

Variables	CTRL-T0	CTRL-T1	*p*-Value
Sample size	30 (100.00)	30 (100.00)	
Female (%)	16 (53.30)	16 (53.30)	
Male (%)	14 (46.70)	14 (46.70)	
Weight (kg)	76.12 ± 19.93	75.19 ± 18.66	0.95
BMI (kg/m^2^)	25.78 ± 5.54	25.46 ± 5.03	0.87

Values are expressed as mean ± standard deviation (M ± SD) for continuous variables, and as number and percentage (n (%)) for categorical variables. Abbreviations: BMI (body mass index).

**Table 5 nutrients-18-00193-t005:** MEDAS item responses and overall scores in ALZ group at T0 and T1.

Questions	Answers	ALZ-T0	ALZ-T1	*p*-Value
Olive oil, main dressing				1.00
	Yes	30 (100.00)	30 (100.00)	
	No	0 (0.00)	0 (0.00)	
Olive oil, ≥4 ts/day				0.20
	Yes	12 (40.00)	18 (60.00)	
	No	18 (60.00)	12 (40.00)	
Vegetables, ≥2 s/day				<0.01 ***
	Yes	7 (23.33)	21 (70.00)	
	No	23 (76.66)	9 (30.00)	
Fruits, ≥3 s/day				0.57
	Yes	10 (33.33)	7 (23.33)	
	No	20 (66.66)	23 (76.66)	
Read meat, ≤1 s/day				0.38
	Yes	24 (80.00)	20 (66.66)	
	No	6 (20.00)	10 (33.33)	
Butter, ≤1 s/day				0.02 *
	Yes	22 (73.33)	29 (96.66)	
	No	8 (26.66)	1 (3.33)	
Sweet beverage, ≤1 s/day				1.00
	Yes	26 (86.66)	27 (90.00)	
	No	4 (13.33)	3 (10.00)	
Red wine, ≥7 s/week				0.18
	Yes	8 (26.66)	3 (10.00)	
	No	22 (73.33)	27 (90.00)	
Legumes, ≥3 s/week				0.78
	Yes	12 (40.00)	10 (33.33)	
	No	18 (60.00)	20 (66.66)	
Fish and seafood, ≥3 s/week				1.00
	Yes	3 (10.00)	3 (10.00)	
	No	27 (90.00)	27 (90.00)	
Sweets, ≤3 s/week				0.78
	Yes	18 (60.00)	20 (66.66)	
	No	12 (40.00)	10 (33.33)	
Nuts, ≥3 s/week				1.00
	Yes	7 (23.33)	8 (26.66)	
	No	23 (76.66)	22 (73.33)	
White meat over red				0.43
	Yes	11 (36.66)	15 (50.00)	
	No	19 (63.33)	15 (50.00)	
“Soffritto”, ≥2 s/week				0.79
	Yes	16 (53.33)	18 (60.00)	
	No	14 (46.66)	12 (40.00)	
Adherence to the MedDiet				0.25
Low		10 (33.33)	4 (13.33)	
Medium		16 (53.33)	21 (70.00)	
High		4 (13.33)	5 (16.66)	

For each MEDAS item, categorical data are presented as the number of participants in each response category and the corresponding percentage within the group, n (%). Abbreviations: c/gg (teaspoons per day), p/gg (portions per day), and p/week (portion per week). Vegetables (daily serving): 1 medium serving = 200 g. Fruit (daily serving): 1 serving = 100–150 g. Red meat/burger/other meat (daily serving): 1 medium serving = 100–150 g. Butter, margarine, or cream (daily serving): 1 medium serving = 12 g. Sugar-sweetened or soft drinks (daily serving): 1 medium serving = 200 mL. Red wine (daily serving): 1 medium serving = 125 mL. Legumes (weekly serving): 1 serving = 150 g. Fish (daily serving): 1 medium serving = 100–150 g. Shellfish (daily serving): 1 medium serving = 200 g. Nuts (weekly serving): 1 serving = 30 g. Soffritto: traditional Mediterranean cooking base prepared with olive oil and finely chopped vegetables (e.g., onion, garlic, celery, carrot). * *p*-value < 0.05; *** *p*-value < 0.0005.

**Table 6 nutrients-18-00193-t006:** MEDAS item responses and overall scores in control group at T0 and T1.

Questions	Answers	CTRL-T0	CTRL-T1	*p*-Value
Olive oil, main dressing				0.90
	Yes	29 (100.00)	28 (100.00)	
	No	0 (0.00)	0 (0.00)	
Olive oil, ≥4 ts/day				0.46
	Yes	17 (58.62)	20 (71.42)	
	No	12 (41.37)	8 (28.57)	
Vegetables, ≥2 s/day				1.00
	Yes	24 (82.75)	24 (85.71)	
	No	5 (17.24)	4 (14.28)	
Fruits, ≥3 s/day				0.16
	Yes	3 (10.34)	8 (28.57)	
	No	26 (89.65)	20 (71.42)	
Read meat, ≤1 s/day				1.00
	Yes	24 (82.75)	23 (82.14)	
	No	5 (17.24)	5 (17.85)	
Butter, ≤1 s/day				0.46
	Yes	25 (86.20)	21 (75.00)	
	No	4 (13.79)	7 (25.00)	
Sweet beverage, ≤1 s/day				0.48
	Yes	24 (82.75)	20 (71.42)	
	No	5 (17.24)	8 (28.57)	
Red wine, ≥7 s/week				1.00
	Yes	4 (13.79)	3 (10.71)	
	No	25 (86.20)	25 (89.28)	
Legumes, ≥3 s/week				0.65
	Yes	17 (58.62)	19 (67.85)	
	No	12 (41.37)	9 (32.14)	
Fish and seafood, ≥3 s/week				0.23
	Yes	13 (44.82)	18 (64.28)	
	No	16 (55.17)	10 (35.71)	
Sweets, ≤3 s/week				1.00
	Yes	22 (75.86)	22 (78.57)	
	No	7 (24.13)	6 (21.42)	
Nuts, ≥3 s/week				0.50
	Yes	14 (48.27)	17 (60.71)	
	No	15 (51.72)	11 (39.28)	
White meat over red				0.42
	Yes	24 (82.75)	26 (92.85)	
	No	5 (17.24)	2 (7.14)	
“Soffritto”, ≥2 s/week				1.00
	Yes	13 (44.82)	13 (46.42)	
	No	16 (55.17)	15 (53.57)	
Adherence to the MedDiet				0.40
Low		4 (13.79)	1 (3.57)	
Medium		14 (48.27)	13 (46.42)	
High		11 (37.93)	14 (50.00)	

For each MEDAS item, categorical data are presented as the number of participants in each response category and the corresponding percentage within the group, n (%). Abbreviations: c/gg (teaspoons per day), p/gg (portions per day), p/week (portion per week). Vegetables (daily serving): 1 medium serving = 200 g. Fruit (daily serving): 1 serving = 100–150 g. Red meat/burger/other meat (daily serving): 1 medium serving = 100–150 g. Butter, margarine, or cream (daily serving): 1 medium serving = 12 g. Sugar-sweetened or soft drinks (daily serving): 1 medium serving = 200 mL. Red wine (daily serving): 1 medium serving = 125 mL. Legumes (weekly serving): 1 serving = 150 g. Fish (daily serving): 1 medium serving = 100–150 g. Shellfish (daily serving): 1 medium serving = 200 g. Nuts (weekly serving): 1 serving = 30 g. Soffritto: traditional Mediterranean cooking base prepared with olive oil and finely chopped vegetables (e.g., onion, garlic, celery, carrot).

**Table 7 nutrients-18-00193-t007:** Percentage composition of gut bacterial species among participants with Alzheimer’s disease.

Species	ALZ-T0	ALZ-T1	*p*-Value
(Bacillota) *Aminipila butyrica*	0.0169 ± 0.0724	0.1180 ± 0.2563	0.0006 **
(Bacillota) *Anaerobutyricum hallii*	0.2176 ± 0.5175	2.7543 ± 2.4665	<0.0001 ***
(Bacillota) *Anaeromassilibacillus senegalensis*	0.0626 ± 0.206	0.1190 ± 0.1173	0.0075 *
(Bacillota) *Anaerotignum faecicola*	0.0225 ± 0.0899	0.0510 ± 0.0903	0.0066 *
(Bacillota) *Anaerotruncus rubiinfantis*	0.0015 ± 0.0061	0.0450 ± 0.1111	0.0008 **
(Bacillota) *Blautia coccoides*	0.0000 ± 0.0000	0.1183 ± 0.2874	0.0025 **
(Bacillota) *Blautia faecicola*	0.0351 ± 0.1261	0.5233 ± 0.8648	0.0001 ***
(Bacillota) *Blautia glucerasea*	0.2360 ± 0.5743	2.249 ± 2.2846	<0.0001 ***
(Bacillota) *Blautia hansenii*	0.0026 ± 0.0103	0.0750 ± 0.1676	0.0028 **
(Bacillota) *Blautia luti*	0.6157 ± 1.2666	5.6243 ± 4.3254	<0.0001 ***
(Bacillota) *Blautia obeum*	2.2803 ± 2.3148	0.5566 ± 0.7914	0.0012 **
(Bacillota) *Blautia schinkii*	0.0003 ± 0.0018	0.0353 ± 0.0829	0.0058 *
(Bacillota) *Blautia stercoris*	0.0253 ± 0.1387	0.1320 ± 0.4666	0.0025 **
(Bacillota) *Butyricicoccus faecihominis*	0.1270 ± 0.3697	0.4150 ± 0.3986	0.0007 **
(Bacillota) *Caproiciproducens galactitolivorans*	0.0041 ± 0.0162	0.0230 ± 0.0329	0.0096 *
(Bacillota) *Christensenella massiliensis*	0.0120 ± 0.062	0.0700 ± 0.1157	0.0007 **
(Bacillota) *Christensenella timonensis*	0.0447 ± 0.2408	0.1460 ± 0.449	0.0038 **
(Bacillota) *Clostridium merdae*	0.0000 ± 0.0000	0.0113 ± 0.0223	0.0057 *
(Bacillota) *Clostridium saudiense*	0.2841 ± 0.7238	0.6713 ± 0.9365	0.0085 *
(Bacillota) *Dorea formicigenerans*	0.7113 ± 0.7779	0.1370 ± 0.4471	0.0010 **
(Bacillota) *Emergencia timonensis*	0.0012 ± 0.0065	0.0356 ± 0.0594	0.0025 **
(Bacillota) *Enterocloster bolteae*	0.0561 ± 0.1959	0.4436 ± 1.0341	0.0005 **
(Bacillota) *Enterocloster clostridioformis*	0.0003 ± 0.0018	0.0590 ± 0.1364	0.0091 *
(Bacillota) *Eubacterium coprostanoligenes*	0.1787 ± 0.4154	1.2096 ± 0.8979	<0.0001 ***
(Bacillota) *Eubacterium oxidoreducens*	0.0051 ± 0.0162	0.2353 ± 0.5857	0.0018 **
(Bacillota) *Agathobacter rectalis*	1.7243 ± 2.7026	0.0000 ± 0.0000	0.0017 **
(Bacillota) *Faecalicatena contorta*	0.0861 ± 0.2281	0.5413 ± 0.5023	<0.0001 ***
(Bacillota) *Faecalicatena orotica*	0.0000 ± 0.0000	0.0106 ± 0.0185	0.0055 *
(Bacillota) *Faecalimonas umbilicata*	0.0286 ± 0.157	0.2830 ± 0.5415	0.0003 ***
(Bacillota) *Flavonifractor plautii*	0.2365 ± 0.4021	0.7066 ± 1.6466	0.0085 *
(Bacillota) *Harryflintia acetispora*	0.0240 ± 0.1277	0.1090 ± 0.2112	0.0066 *
(Bacillota) *Holdemanella biformis*	0.3570 ± 1.1373	1.1343 ± 2.4272	0.0052 *
(Bacillota) *Holdemania filiformis*	0.0000 ± 0.0000	0.0150 ± 0.0293	0.0055 *
(Bacillota) *Intestinibacter bartlettii*	0.2727 ± 0.4788	0.7296 ± 0.8738	0.0037 **
(Bacillota) *Kineothrix alysoides*	0.0488 ± 0.2344	0.3083 ± 0.5122	0.0052
(Bacillota) *Lachnoclostridium pacaense*	0.0604 ± 0.2815	0.2306 ± 0.4917	0.0090 *
(Bacillota) *Lachnospira eligens*	0.0717 ± 0.2344	0.5146 ± 0.8247	0.0031 **
(Bacillota) *Lacrimispora amygdalina*	0.0193 ± 0.082	0.1120 ± 0.2405	0.0060 *
(Bacillota) *Lacrimispora sphenoides*	0.0069 ± 0.0381	0.0833 ± 0.1425	0.0069 *
(Bacillota) *Lactobacillus rogosae*	0.0076 ± 0.0419	0.1190 ± 0.1788	0.0029 **
(Bacillota) *Longibaculum muris*	0.0380 ± 0.2081	0.6440 ± 1.3071	0.0001 ***
(Bacillota) *Longicatena caecimuris*	0.0014 ± 0.008	0.0600 ± 0.1284	0.0047 **
(Bacillota) *Mediterraneibacter faecis*	0.1470 ± 0.4627	0.8993 ± 1.041	0.0001 ***
(Bacillota) *Mediterraneibacter glycyrrhizinilyticus*	0.0013 ± 0.0073	0.0513 ± 0.1467	0.0059
(Bacillota) *Monoglobus pectinilyticus*	0.0600 ± 0.1581	0.4556 ± 1.2353	0.0019 **
(Bacillota) *Oscillibacter ruminantium*	0.0970 ± 0.2994	0.5796 ± 0.8144	0.0013 **
(Bacillota) *Oscillibacter valericigenes*	0.5017 ± 1.4513	1.677 ± 1.491	0.0010 **
(Bacillota) *Phascolarctobacterium faecium*	0.1552 ± 0.5021	0.4840 ± 0.9071	0.0013 **
(Bacillota) *Phocea massiliensis*	0.0485 ± 0.208	0.1773 ± 0.3743	0.0094 *
(Bacillota) *Pseudoflavonifractor phocaeensis*	0.0420 ± 0.1057	0.1943 ± 0.3102	0.0020 **
(Bacillota) *Romboutsia timonensis*	0.1853 ± 0.6888	2.0496 ± 3.3988	0.0001 ***
(Bacillota) *Ruminococcus albus*	0.1671 ± 0.6239	1.056 ± 1.356	0.0012 **
(Bacillota) *Ruminococcus bicirculans*	0.2320 ± 0.4356	0.0000 ± 0.0000	0.0025 **
(Bacillota) *Ruminococcus callidus*	0.0356 ± 0.1156	0.7000 ± 1.2887	0.0007 **
(Bacillota) *Mediterraneibacter gnavus*	0.2150 ± 0.5503	0.0000 ± 0.0000	0.0059 *
(Bacillota) *Mediterraneibacter torques*	1.4363 ± 2.2829	0.0000 ± 0.0000	0.0001 ***
(Bacillota) *Sporobacter termitidis*	0.0809 ± 0.2343	0.3716 ± 0.4107	0.0024 **
(Bacillota) *Streptococcus troglodytae*	0.0003 ± 0.0018	0.0393 ± 0.0906	0.0016 **

Values are expressed as the mean ± standard deviation (M ± SD) for continuous variables. In parentheses, the phylum is indicated; species names follow binomial nomenclature (genus and specific epithet). * *p*-value < 0.05; ** *p*-value < 0.005; *** *p*-value < 0.0005.

**Table 8 nutrients-18-00193-t008:** Percentage composition of gut bacterial species in control group.

Species	CTRL-T0	CTRL-T1	*p*-Value
(Bacillota) *Anaeromassilibacillus senegalensis*	0.14 ± 0.15	0.15 ± 0.35	0.27
(Bacillota) *Anaerostipes hadrus*	2.00 ± 1.60	2.09 ± 2.15	0.89
(Bacillota) *Blautia faecicola*	0.34 ± 0.72	0.49 ± 0.86	0.45
(Bacillota) *Blautia glucerasea*	1.75 ± 2.86	1.46 ± 2.12	0.81
(Bacillota) *Blautia luti*	4.92 ± 5.13	4.95 ± 5.78	0.79
(Bacillota) *Blautia obeum*	0.63 ± 0.87	1.48 ± 2.53	0.22
(Bacillota) *Blautia wexlerae*	0.27 ± 0.68	0.22 ± 0.36	0.19
(Bacillota) *Coprococcus catus*	0.20 ± 0.30	0.14 ± 0.18	0.74
(Bacillota) *Dorea longicatena*	2.04 ± 1.89	1.86 ± 1.52	0.97
(Bacillota) *Faecalibacterium prausnitzii*	1.78 ± 5.39	9.04 ± 6.25	0.30
(Bacillota) *Flavonifractor plautii*	0.29 ± 0.33	0.29 ± 0.50	0.37
(Bacillota) *Fusicatenibacter saccharivorans*	2.26 ± 2.04	1.69 ± 1.80	0.13
(Bacillota) *Monoglobus pectinilyticus*	0.34 ± 0.52	0.33 ± 0.58	0.58
(Bacillota) *Oscillibacter valericigenes*	1.05 ± 1.60	1.18 ± 2.06	0.98
(Bacillota) *Romboutsia timonensis*	1.02 ± 1.51	1.15 ± 1.45	0.73
(Bacillota) *Roseburia faecis*	2.12 ± 2.50	1.58 ± 1.98	0.36
(Bacillota) *Roseburia Hominis*	0.52 ± 0.65	0.24 ± 0.38	0.14
(Bacillota) *Roseburia inulinivorans*	0.41 ± 0.65	0.61 ± 1.32	0.41
(Bacillota) *Sporobacter termitidis*	0.45 ± 0.81	0.22 ± 0.43	0.40
(Bacillota) *Streptococcus salivarius*	0.21 ± 0.85	0.22 ± 0.41	0.12
(Bacillota) *Streptococcus thermophilus*	0.58 ± 1.06	0.39 ± 1.05	0.19

Values are expressed as the mean ± standard deviation (M ± SD) for continuous variables. In parentheses, the phylum is indicated; species names follow binomial nomenclature (genus and specific epithet).

**Table 9 nutrients-18-00193-t009:** Δ relative abundance (T1-T0) of gut microbiota genera in ALZ and CTRL patients.

Genera	Median Δ ALZ	Median Δ CTRL	*p*-Value
(Pseudomonadota) *Gemmiger* sp.	−0.0500	0.0000	0.0006 ***
(Bacillota) *Subdoligranulum* sp.	−0.3500	0.0000	0.0001 ***
(Bacillota) *Anaerobutyricum* sp.	2.2000	0.0000	<0.00001 ***
(Bacillota) *Romboutsia* sp.	0.2500	0.0000	0.0072 **
(Bacillota) *Faecalicatena* sp.	0.4000	0.0000	0.0063 **
(Actinomycetota) *Arthrobacter* sp.	0.0000	0.0000	0.0068 **
(Bacillota) *Sporobacter* sp.	0.0000	0.0000	0.0073 **
(Bacillota) *Mediterraneibacter* sp.	0.1500	0.0000	0.0024 **
(Bacillota) *Blautia* sp.	3.7500	0.1500	0.0459 *
(Actinomycetota) *Bifidobacterium* sp.	−1.9500	−0.35	0.0475 *
(Bacillota) *Clostridium* sp.	−0.8500	0.0000	0.0422 *
(Actinomycetota) *Collinsella* sp.	−0.8000	0.0000	0.0426 *
(Bacillota) *Oscillibacter* sp.	1.2500	0.0000	0.0267 *
(Bacillota) *Erysipelatoclostridium* sp.	0.0000	0.0000	0.0144 *
(Bacillota) *Veillonella* sp.	0.0000	0.0000	0.0419 *
(Pseudomonadota) *Escherichia* sp.	0.0000	0.0000	0.048 *
(Bacillota) *Lachnospira* sp.	0.0000	0.0000	0.0121 *
(Bacillota) *Holdemanella* sp.	0.0000	0.0000	0.0448 *
(Bacillota) *Limosilactobacillus* sp.	0.0000	0.0000	0.0419 *
(Bacillota) *Christensenella* sp.	0.0000	0.0000	0.0111 *
(Methanobacteriota) *Methanobrevibacter* sp.	0.0000	0.0000	0.0248 *
(Actinomycetota) *Actinomyces* sp.	0.0000	0.0000	0.0216 *

In parentheses, the phylum is indicated; the following name refers to the genus. * *p*-value < 0.05; ** *p*-value < 0.005; *** *p*-value < 0.0005.

**Table 10 nutrients-18-00193-t010:** The log2 fold change in the relative abundance of gut microbiota genera compared between ALZ and CTRL patients.

Genera	Median Δ ALZ	Median Δ CTRL	log_2_FC	*p*-Value
(Bacillota) *Blautia* sp.	3.75	0.15	4.64	0.04 *
(Actinomycetota) *Bifidobacterium* sp.	−1.95	−0.35	2.47	0.04 *
(Bacillota) *Romboutsia* sp.	0.25	0.00	17.93	<0.01 **
(Bacillota) *Oscillibacter* sp.	1.25	0.00	20.25	0.02 *
(Bacillota) *Anaerobutyricum* sp.	2.20	0.00	21.06	<0.01 ***
(Bacillota) *Faecalicatena* sp.	0.04	0.00	18.60	<0.01 **
(Bacillota) *Mediterraneibacter* sp.	0.15	0.00	17.19	<0.01 **

In parentheses, the phylum is indicated; the following name refers to the genus. * *p*-value < 0.05; ** *p*-value < 0.005; *** *p*-value < 0.0005.

## Data Availability

The original contributions presented in this study are included in the article/Appendix A. Further inquiries can be directed to the corresponding authors.

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
