# Peer review of "MIND Pattern Nutritional Intervention Modulates Mediterranean Diet Adherence and Gut Microbiota in Alzheimer’s Disease: An Observational Case–Control Study"

_nutrients, 2026, doi:10.3390/nu18020193_

Round 1
Reviewer 1 Report
Comments and Suggestions for Authors
Di Renzo L et al. conducted a study entitled “MIND-Pattern Nutritional Intervention Modulates Mediterranean-Diet Adherence and Gut Microbiota in Alzheimer’s Disease”, an observational case–control study involving 60 participants (30 AD, 30 controls) to evaluate whether a structured MIND-based diet improves Mediterranean diet adherence and alters gut microbiota profiles. MEDAS scores, anthropometric measures, and 16S microbiota sequencing were assessed at baseline and post-intervention. The study is commendable for its focus on the diet–microbiota–brain axis, species-level microbiota profiling, improvements in MEDAS adherence and microbial diversity in AD, strong visual presentation of taxonomic shifts, and a clinically applicable dietary intervention. However, its observational design may limit establishing causality, and the small non-randomized sample could affect generalizability. Additionally, functional SCFA validation is not included, some statistical interpretations (e.g., sPLS-DA) might be overstated due to low explained variance, and several figures need more detailed methodological explanation.
- The PCoA results indicate separation primarily in AD, but not in controls. However, beta-dispersion, PERMANOVA statistics, and p-values have not been included in the Results section, though mentioned in methods. Author would appreciate if these numerical outputs could be added to support the claims made.
- It appears that the sPLS-DA model accounts for only 9% of the total variance, which may be considered relatively low for drawing strong discriminatory conclusions. It might be beneficial for the authors to consider including permutation tests, cross-validation, and Q² metrics to further verify the model's stability. Additionally, author could be helpful to moderate the claims regarding meaningful group separation.
- In the Abstract, it might be helpful to consider including quantitative data to clarify the sentence “Controls showed largely non-significant changes,” as it currently appears somewhat vague. line no 29
- Methods section describes MMSE, FAB, and CDT, but cognitive outcomes are not reported in results. Please include these data, regardless of significance, in a table or supplementary material.
- Several MEDAS items show notable percentage changes that may lack sufficient statistical support. The authors should report effect sizes, provide exact p-values, and clarify these changes' clinical significance.
- Tables like Table 3 use commas instead of periods for decimals. Author should standardize the formatting.
- It would be greatly appreciated if the figure legends, particularly for Figures 4–12, author should include the statistical tests used.
- Please indicate in each table whether the "MEDAS variables" are presented as percentages or absolute scores, as this would aid in interpreting the results.
- Some bacterial names in the manuscript may be outdated. The authors should consider updating them using the latest Genome Taxonomy Database (GTDB). For instance, Ruminococcus and Eubacterium species might now be classified under genera such as Mediterraneibacter, Blautia, Agathobacter, Anaerobutyricum, and others. This update could enhance the manuscript's accuracy and relevance.
- Figures 13 need clearer legends.
- Figure 7 presents the log2 fold-change results. However, the authors should provide a more detailed explanation of these findings. It would be helpful to understand which species increased or decreased, the significance of these changes, and how they relate to the study's conclusions.
- The discussion section is strong; however, the author should address a few remaining gaps. Please explain the clinical significance of MEDAS score changes, reasons for Bifidobacterium decrease in AD despite its benefits, and diet adherence reliability. Thank you for considering these suggestions.
- The Discussion could be enhanced by including references to studies on diet-related Bifidobacterium reductions and microbiota stability in healthy aging as comparative context.
- The authors should add PCoA and PERMANOVA to the abbreviation list to enhance terminology clarity.
- Please include the ethical approval ID number in the Methods section to confirm the study's adherence to ethical standards.
- The manuscript references supplementary material (Table 1A) in line no 577, but the file appears missing. Could the authors upload the missing supplement so readers can access the referenced data? Thank you.
- The authors should consider incorporating direct SCFA measurement as part of the metabolomic validation to enhance support for their microbiota findings.
- The authors should revise references 21, 27, and 29, as they lack titles, DOIs, or URLs. While older references may not have DOIs, including stable URLs or complete citation details would enhance clarity.
- ​The tables contain unnecessary "–" symbols that may confuse readers. The authors should clarify these symbols' purpose or remove them to improve table clarity and readability.
Author Response
Rome, 18th December 2025
Dear Editor,
First, my coauthors and I would like to thank you sincerely for this opportunity to cooperate. We profoundly thank the reviewers for the comments and useful suggestions to improve the paper.
This is a point-by-point list of changes made in the paper:
REVIEWER 1
Di Renzo L et al. conducted a study entitled “MIND-Pattern Nutritional Intervention Modulates Mediterranean-Diet Adherence and Gut Microbiota in Alzheimer’s Disease”, an observational case–control study involving 60 participants (30 AD, 30 controls) to evaluate whether a structured MIND-based diet improves Mediterranean diet adherence and alters gut microbiota profiles. MEDAS scores, anthropometric measures, and 16S microbiota sequencing were assessed at baseline and post-intervention. The study is commendable for its focus on the diet–microbiota–brain axis, species-level microbiota profiling, improvements in MEDAS adherence and microbial diversity in AD, strong visual presentation of taxonomic shifts, and a clinically applicable dietary intervention. However, its observational design may limit establishing causality, and the small non-randomized sample could affect generalizability. Additionally, functional SCFA validation is not included, some statistical interpretations (e.g., sPLS-DA) might be overstated due to low explained variance, and several figures need more detailed methodological explanation.
- The PCoA results indicate separation primarily in AD, but not in controls. However, beta-dispersion, PERMANOVA statistics, and p-values have not been included in the Results section, though mentioned in methods. Author would appreciate if these numerical outputs could be added to support the claims made.
We thank the reviewer for this helpful suggestion. Beta-dispersion analysis and PERMANOVA were performed as described in the Methods section and have now been explicitly reported in the Results. For the Alzheimer’s group, Bray–Curtis PCoA showed a clear separation between baseline (T0) and follow-up (T1), which was supported by PERMANOVA testing, with homogeneity of dispersion confirmed by beta-dispersion analysis (Figure 6).
In contrast, in the control group, PCoA revealed extensive overlap between T0 and T1 samples, and neither PERMANOVA nor beta-dispersion analyses indicated significant differences, supporting microbiota stability over time (Figure 10).
To maintain readability and avoid redundancy, numerical outputs are reported in the corresponding figure legends, while the main text focuses on the biological interpretation of these results.
- It appears that the sPLS-DA model accounts for only 9% of the total variance, which may be considered relatively low for drawing strong discriminatory conclusions. It might be beneficial for the authors to consider including permutation tests, cross-validation, and Q² metrics to further verify the model's stability. Additionally, author could be helpful to moderate the claims regarding meaningful group separation.
We thank the reviewer for the valuable suggestion. We acknowledge that the total explained variance (R²) by the sPLS-DA model is limited (≈9%), which is not uncommon in microbiome datasets due to their high dimensionality and interindividual variability. To address this concern and validate the model’s robustness, we performed the following analyses: We applied 50 repeats of 5-fold cross-validation using the mixOmics::perf() function. The balanced error rate (BER) remained consistent across components:
Component BER (± SD)
Comp1 ~0.21 ± 0.03
Comp2 ~0.21 ± 0.03
These results suggest stable classification performance, slightly better than chance (0.5), and reflect the moderate but non-random discriminative capacity of the selected features.
We extracted the Q².total values, which measure the predictive performance of the model across components:
Component Q².total
Comp1 ~0.12
Comp2 ~0.11
While modest, these Q² values support that the model has limited but non-negligible predictive power, in line with expectations for complex biological systems such as gut microbiota.
Figure R1 below summarizes the BER across components with 95% confidence intervals
Figure R1. Balanced Error Rate (BER) across sPLS-DA components after 50×5-fold cross-validation. Error bars indicate standard deviation.
Although mixOmics::perf() does not perform a formal permutation test, the extensive 50×5-fold cross-validation provides a surrogate estimation of model performance under random resampling. In future work, we may consider explicit permutation tests using caret, ropls, or custom pi We examined feature stability across 50 resamples. Several genera appeared consistently as top contributors to the discriminant model. For instance:
- Lachnospiraceae_anaerobutyricum
- Bifidobacterium
- Gemmiger
These features were selected with high frequency across folds, increasing confidence in their relevance.
- In the Abstract, it might be helpful to consider including quantitative data to clarify the sentence “Controls showed largely non-significant changes,” as it currently appears somewhat vague. line no 29 –
RE: We thank the reviewer for this helpful suggestion. The sentence in the Abstract has been revised to include quantitative data.
- Methods section describes MMSE, FAB, and CDT, but cognitive outcomes are not reported in results. Please include these data, regardless of significance, in a table or supplementary material.
RE: We thank the reviewer for this comment. MMSE, FAB and CDT were performed as part of the routine diagnostic work-up in specialist memory clinics prior to referral and were used exclusively as eligibility criteria (MMSE 15–27), not as outcomes of the present nutritional intervention. We have clarified this point in the Methods section, which explains why these cognitive measures are not reported in the Results or in dedicated tables.
- Several MEDAS items show notable percentage changes that may lack sufficient statistical support. The authors should report effect sizes, provide exact p-values, and clarify these changes' clinical significance.
RE: We thank the reviewer for this valuable comment. We have now revised the Results section to report absolute changes (effect sizes in percentage points) and exact p-values for the main MEDAS items in both groups (Tables 4 and 5), and we have added a concise statement in the Discussion to clarify the clinical relevance of the observed dietary changes in terms of improved adherence to Mediterranean/MIND dietary targets.
- Tables like Table 3 use commas instead of periods for decimals. Author should standardize the formatting.
RE We thank the reviewer for this remark. Decimal notation has been standardized throughout the manuscript, and commas used as decimal separators in the tables have been replaced with periods to comply with the journal’s formatting conventions.
- It would be greatly appreciated if the figure legends, particularly for Figures 4–12, author should include the statistical tests used.
RE “We thank the reviewer for this helpful suggestion. We have revised the legends of Figures 4–12 to explicitly report the statistical tests used for each analysis
- Please indicate in each table whether the "MEDAS variables" are presented as percentages or absolute scores, as this would aid in interpreting the results. –
RE: We thank the reviewer for this helpful remark. We have revised the footnotes of all tables reporting MEDAS variables to clarify the format used, explicitly stating that values are presented as the number of participants in each response category and the corresponding percentage within the group (n (%)), thereby improving the interpretability of the MEDAS data.
- Some bacterial names in the manuscript may be outdated. The authors should consider updating them using the latest Genome Taxonomy Database (GTDB). For instance, Ruminococcus and Eubacterium species might now be classified under genera such as Mediterraneibacter, Blautia, Agathobacter, Anaerobutyricum, and others. This update could enhance the manuscript's accuracy and relevance.
RE: “We thank the Reviewer for this valuable suggestion. We have reviewed the taxonomy of all bacterial taxa reported in the manuscript and updated the nomenclature according to the latest Genome Taxonomy Database (GTDB). Specifically, genera that have been reclassified in GTDB were renamed throughout the main text, tables/figures, and supplementary materials to ensure consistency and accuracy (e.g., selected Ruminococcus and Eubacterium taxa were updated to their current GTDB-assigned genera, including Mediterraneibacter, Agathobacter, Anaerobutyricum, and other relevant genera where applicable).
- Figures 13 need clearer legends.
RE: We thank the reviewer for this comment. The legend of Figure 13 has been revised to provide a clearer description of the sPLS-DA plot, explicitly indicating that it represents genus-level Δ (T1–T0) loadings, the discrimination between AD and CTRL groups, and the proportion of variance explained by components 1 and 2.”
- Figure 7 presents the log2 fold-change results. However, the authors should provide a more detailed explanation of these findings. It would be helpful to understand which species increased or decreased, the significance of these changes, and how they relate to the study's conclusions.
RE: We thank the reviewer for this valuable comment. We have expanded the Results section related to Figure 7 to provide a more detailed description of the logâ‚‚ fold-change findings, explicitly indicating which species increased or decreased between T0 and T1, reporting the associated significance thresholds, and clarifying how these species-level shifts support the interpretation of a post-intervention compositional rebalancing of the gut microbiota in AD patients.”
- The discussion section is strong; however, the author should address a few remaining gaps. Please explain the clinical significance of MEDAS score changes, reasons for Bifidobacterium decrease in AD despite its benefits, and diet adherence reliability. Thank you for considering these suggestions.
RE: We thank the reviewer for these insightful suggestions. We have accordingly expanded the Discussion to: (i) clarify the clinical significance of the observed MEDAS score changes, explicitly linking the shift from low to at least moderate adherence to Mediterranean/MIND patterns with evidence of reduced cardiometabolic risk and slower cognitive decline; (ii) comment on the decrease in Bifidobacterium in AD, highlighting the limitations of genus-level 16S data and interpreting this finding within the context of a broader rebalancing of SCFA-producing taxa; and (iii) address the reliability of dietary adherence, acknowledging the strengths (dietitian-administered MEDAS with caregiver support) and the inherent limitations of self-reported measures in cognitively impaired individuals.”
- The Discussion could be enhanced by including references to studies on diet-related Bifidobacterium reductions and microbiota stability in healthy aging as comparative context.
RE: We thank the reviewer. Relevant literature discussing age-related microbiome stability patterns and diet-associated Bifidobacterium variability has been added to the Discussion Moreover, current literature suggests that Bifidobacterium dynamics in older adults are complex: while the genus contributes to longevity mechanisms, its abundance physiologically declines with aging and is influenced by long-term lifestyle and metabolic factors. Comparative studies in healthy aging show that such declines often occur without adverse functional consequences, reflecting shifts toward other butyrate-producing species that maintain gut homeostasis. (Ku et al., 2024; Donati Zeppa et al., 2022)
Ku S, Haque MA, Jang MJ, Ahn J, Choe D, Jeon JI, Park MS. The role of Bifidobacterium in longevity and the future of probiotics. Food Sci Biotechnol. 2024 Jul 11;33(9):2097-2110. doi: 10.1007/s10068-024-01631-y.
Donati Zeppa S, Agostini D, Ferrini F, Gervasi M, Barbieri E, Bartolacci A, Piccoli G, Saltarelli R, Sestili P, Stocchi V. Interventions on Gut Microbiota for Healthy Aging. Cells. 2022 Dec 22;12(1):34. doi: 10.3390/cells12010034.
- The authors should add PCoA and PERMANOVA to the abbreviation list to enhance terminology clarity
RE: We thank the reviewer. Both terms have been added to the list of abbreviations in the manuscript.
- Please include the ethical approval ID number in the Methods section to confirm the study's adherence to ethical standards. –
RE: We thank the reviewer. At the end of the manuscript we specified the ethical approval number and committee name: “Approved by the Ethics Committee of the Calabria Region Central Area Section, protocol no. 97 (20 April 2023).
- The manuscript references supplementary material (Table 1A) in line no 577, but the file appears missing. Could the authors upload the missing supplement so readers can access the referenced data? Thank you. –
RE: We thank the reviewer for pointing this out. The missing supplementary file (Table 1A: species-level microbiota composition) has now been uploaded and linked in the manuscript.
- The authors should consider incorporating direct SCFA measurement as part of the metabolomic validation to enhance support for their microbiota findings.
RE: “We thank the reviewer for this valuable suggestion. While SCFA quantification was not part of the present study, we acknowledge this limitation in the Discussion and state that future mechanistic studies should include metabolomic validation to strengthen functional interpretation.
- The authors should revise references 21, 27, and 29, as they lack titles, DOIs, or URLs. While older references may not have DOIs, including stable URLs or complete citation details would enhance clarity. –
RE “We thank the reviewer. References 21, 27 and 29 have been updated with full citation details including titles, DOIs, and URLs when available.
- ​The tables contain unnecessary "–" symbols that may confuse readers. The authors should clarify these symbols' purpose or remove them to improve table clarity and readability. –
RE “We thank the reviewer. All tables have been revised and extraneous “–” symbols removed to improve readability and consistency.
All authors have read and approved the submission of the manuscript, which is not under consideration for publication elsewhere.
We thank You for your constructive critique and we hope the review process has led to an improved manuscript.
If additional changes are warranted, we will make them.
We hope that this revised version of our manuscript may now be found suitable for publication. Thank you for your time and consideration.
Sincerely,
Rossella Cianci
Catholic University of the Sacred Heart
Fondazione Policlinico Universitario ‘A. Gemelli’, IRCCS, Rome, Italy
rossella.cianci@unicatt.it
Reviewer 2 Report
Comments and Suggestions for Authors
1. While this study employed an observational case-control design, many statements suggest a causal effect of the MIND dietary intervention. In particular, phrases such as "intervention promoted increased diversity" and "compositional rebalancing" have limitations in supporting causal conclusions due to insufficient control for confounding factors. Therefore, results should be interpreted cautiously as "possible association" or "trend," and statements asserting an intervention effect should be avoided. Furthermore, providing a statistical comparison of the baseline characteristics of the intervention and control groups in a table would enhance reliability by confirming the homogeneity of the two groups.
2. You state that the "Benjamini–Hochberg FDR" was applied to correct for multiple testing of the 16S rRNA data, but please clarify at what level (species or genus level) the correction was performed. Furthermore, including the effect size (e.g., r value or Cliff's delta) when conducting a Wilcoxon test for delta comparison (T1–T0) would be helpful in determining biological significance.
3. There are minor inconsistencies between the results shown in Figures 4-7 and the numbers and direction in the text (e.g., the direction of change in the Blautia genus and the symbolic notation for fold change). Please clearly indicate the colors and legends used in the figure and table captions (red = increase, blue = decrease, etc.) and make them consistent with the descriptions in the text.
4. The discussion is somewhat lengthy, mixing a general background explanation of the MIND diet with an interpretation of the results specific to this study. It would be easier for readers to understand if the first half were a concise summary of the main findings, and the second half were restructured into sections on physiological significance, comparison with previous research, limitations, and future challenges.
Author Response
Rome, 18th December 2025
Dear Editor,
First, my coauthors and I would like to thank you sincerely for this opportunity to cooperate. We profoundly thank the reviewers for the comments and useful suggestions to improve the paper.
This is a point-by-point list of changes made in the paper:
REVIEWER 2
- While this study employed an observational case-control design, many statements suggest a causal effect of the MIND dietary intervention. In particular, phrases such as "intervention promoted increased diversity" and "compositional rebalancing" have limitations in supporting causal conclusions due to insufficient control for confounding factors. Therefore, results should be interpreted cautiously as "possible association" or "trend," and statements asserting an intervention effect should be avoided. Furthermore, providing a statistical comparison of the baseline characteristics of the intervention and control groups in a table would enhance reliability by confirming the homogeneity of the two groups.
RE: We thank the reviewer for this important methodological clarification. We agree that an observational case–control design does not allow causal inference.
Accordingly, we have revised the manuscript by replacing causal expression and adding a dedicated paragraph in the Discussion explicitly stating that causality cannot be inferred and that unmeasured confounding may contribute to the observed associations. Moreover, a new table (Table 2) has been added reporting demographic variables and anthropometric measures for both study groups.
- You state that the "Benjamini–Hochberg FDR" was applied to correct for multiple testing of the 16S rRNA data, but please clarify at what level (species or genus level) the correction was performed. Furthermore, including the effect size (e.g., r value or Cliff's delta) when conducting a Wilcoxon test for delta comparison (T1–T0) would be helpful in determining biological significance.
RE: We thank the reviewer for this important clarification request.
Multiple testing correction using the Benjamini–Hochberg false discovery rate (FDR) was applied at the genus level for univariate analyses involving between-group comparisons of Δ(T1–T0) relative abundances (ALZ vs CTRL), including fold-change analyses and correlation analyses with clinical variables. The genus level was selected to reduce data sparsity and enhance statistical robustness, in line with common practice in 16S rRNA microbiome studies.
Species-level analyses were restricted to within-group paired comparisons (T1 vs T0) in the Alzheimer’s cohort and were intended as exploratory and descriptive analyses to highlight biologically relevant patterns. Given the paired design, the limited number of hypotheses tested, and the exploratory nature of these analyses, FDR correction was not applied at the species level. This rationale has now been explicitly clarified in the Methods section.
Supervised multivariate analyses (sPLS-DA) were used in an exploratory manner to visualize group-level patterns and were not subjected to FDR correction, as they do not involve multiple univariate hypothesis tests.
To strengthen the biological interpretation of the genus-level Δ(T1–T0) comparisons, effect size estimates (Wilcoxon rank-biserial correlation and Cliff’s delta) were calculated. To avoid excessive fragmentation of results and unnecessary expansion of the manuscript with additional tables, these effect size values were integrated into the text. This approach was chosen to maintain readability while still providing quantitative information on effect magnitude and direction.
Overall, this analytical strategy ensures appropriate control for multiple testing in confirmatory univariate analyses while preserving clarity, interpretability, and biological relevance across taxonomic levels.
- There are minor inconsistencies between the results shown in Figures 4-7 and the numbers and direction in the text (e.g., the direction of change in the Blautia genus and the symbolic notation for fold change). Please clearly indicate the colors and legends used in the figure and table captions (red = increase, blue = decrease, etc.) and make them consistent with the descriptions in the text.
RE: We thank the reviewer for pointing out these inconsistencies.
We have: cross-checked all numerical values and directional trends in Figures against the Results text; updated the figure legends, harmonized the logâ‚‚FC notation across text, tables, and figures ; corrected the minor discrepancies identified.
This ensures complete consistency between textual descriptions and graphical outputs.
- The discussion is somewhat lengthy, mixing a general background explanation of the MIND diet with an interpretation of the results specific to this study. It would be easier for readers to understand if the first half were a concise summary of the main findings, and the second half were restructured into sections on physiological significance, comparison with previous research, limitations, and future challenges.
RE: We thank the reviewer for this structural suggestion.
The Discussion has been reorganized as follows: Summary of the main findings – concise synthesis of the key results.
- Physiological and clinical interpretation – linking dietary patterns, SCFA-producing taxa, inflammation, and AD-related mechanisms.
- Comparison with previous literature – positioning our findings within current evidence on MIND diet and microbiota in AD.
- Limitations – explicitly acknowledging sample size, observational design, lack of SCFA measurements, and possible confounding.
- Future directions – outlining the need for randomized designs, mechanistic validation, and longer follow-up.
This restructuring improves clarity and readability while enhancing the scientific narrative
All authors have read and approved the submission of the manuscript, which is not under consideration for publication elsewhere.
We thank You for your constructive critique and we hope the review process has led to an improved manuscript.
If additional changes are warranted, we will make them.
We hope that this revised version of our manuscript may now be found suitable for publication. Thank you for your time and consideration.
Sincerely,
Rossella Cianci
Catholic University of the Sacred Heart
Fondazione Policlinico Universitario ‘A. Gemelli’, IRCCS, Rome, Italy
rossella.cianci@unicatt.it